# UNDERSTANDING AND IMPROVING ADVERSARIAL ATTACKS ON LATENT DIFFUSION MODEL

## ABSTRACT

Latent Diffusion Model (LDM) achieves state-of-the-art performances in image generation yet raising copyright and privacy concerns. Adversarial attacks on LDM are then born to protect unauthorized images from being used in LDM-driven few-shot generation. However, these attacks suffer from moderate performance and excessive computational cost, especially in GPU memory. In this paper, we propose an effective adversarial attack on LDM that shows superior performance against state-of-the-art few-shot generation pipeline of LDM, for example, LoRA. We implement the attack with memory efficiency by introducing several mechanisms and decrease the memory cost of the attack to less than 6GB, which allows individual users to run the attack on a majority of consumer GPUs. Our proposed attack can be a practical tool for people facing the copyright and privacy risk brought by LDM to protect themselves.

## 1 INTRODUCTION

Diffusion models (Sohl-Dickstein et al., 2015; Song & Ermon, 2019; Ho et al., 2020; Song et al., 2020) have long held the promise of producing fine-grained content that could resemble real data. Recently, Latent Diffusion Model (LDM) (Rombach et al., 2022; Podell et al., 2023), a variant of diffusion models, showcased unprecedented capabilities in image generation tasks. LDM's prowess in few-shot generation—generating data with few-shot reference data—has pushed the state-of-the-art performance forward by a significant margin and sparked a craze for AI-generated art (Meng et al., 2021; Gal et al., 2022; Ruiz et al., 2023; Roich et al., 2022; Zhang & Agrawala, 2023).

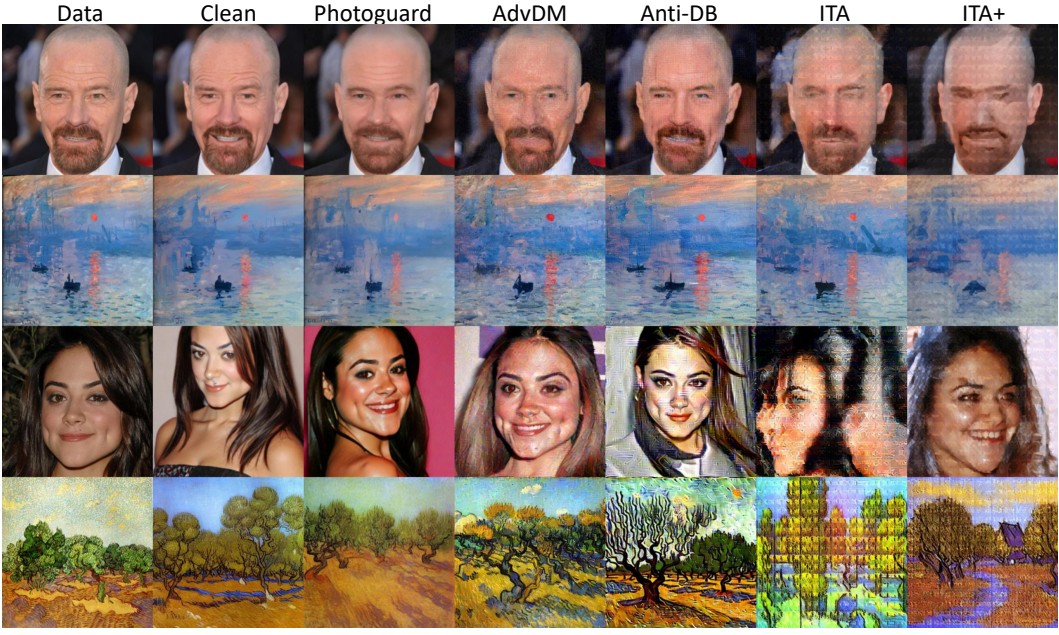

Figure 1: Comparison of outputs of SDEdit (two top rows) and LoRA (two bottom rows) under different attacks. The adversarial budget is $4/255$.

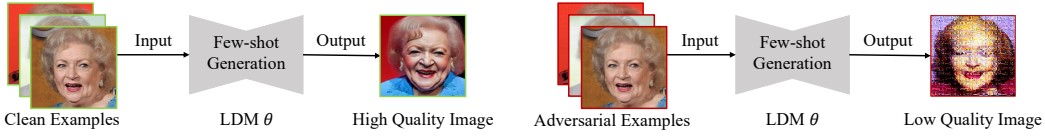

Figure 2: Few-shot generation based on adversarial examples outputs low-quality images.

While the opportunities presented by LDM are immense, the implications of its power are a double-edged sword. Malicious individuals leverage LDM-driven few-shot generation to copy artworks without authorization (Fan et al., 2023) and create fake not-suitable-for-work photos with personal figures (Wang et al., 2023b). Such malevolent applications of LDM threaten the sanctity of personal data and intellectual property.

Recognizing the need, adversarial attacks on LDM were born as countermeasures (Salman et al., 2023; Liang et al., 2023; Shan et al., 2023; Van Le et al., 2023). These attacks add human-invisible perturbations to the real image and transfer it to an adversarial example, making it unusable in LDM-driven few-shot generation. Applications based on these adversarial attacks (Liang & Wu, 2023; Shan et al., 2023) serve as a tool to protect personal images from being used as reference data for LDM-driven few-shot generation.

However, existing adversarial attacks on LDM suffer from moderate effectiveness. Faced with state-of-the-art few-shot generation pipelines, for example, LoRA Hu et al. (2021), these attacks cannot protect the content of images from being learned by LDM and thus taken as reference in malicious image synthesis. Additionally, their requirements for GPU memory also deter normal people who have no access to advanced GPUs from using them. For these two reasons, current adversarial attacks have not yet moved beyond the academic realm to become a practical tool.

In this paper, we improve the adversarial attack on LDM from the aforementioned two aspects. First, we improve its effectiveness against state-of-the-art LLM-driven few-shot generation methods. We do this by designing a new targeted objective function for the attack. Furthermore, we introduce three techniques in memory efficiency in our implementation of the attack to help decrease its GPU memory cost to less than 6GB. We evaluate our attack and existing adversarial attacks on LDM with two mainstream LDM-driven few-shot generation methods, SDEdit (Meng et al., 2021) and LoRA (Hu et al., 2021). Experiments show that our attack outperforms existing adversarial attacks on LDM in both effectiveness and memory efficiency.

**Contributions** Our contributions are two-fold. First, we propose a novel adversarial attack on LDM with visible improvement in effectiveness against both SDEdit and LoRA. Second, we implement three mechanisms to improve the memory efficiency of our attack, which is transferable to other attacks. Both contributions focuses on bottlenecks of current adversarial attacks on LDM.

**Impact** Our attack can serve as a highly usable tool for ordinary people to protect their personal images, including artworks and portraits, from being used as reference data in malicious image synthesis supported by LDM-driven few-shot generation. This is a growing public concern because the popularity of open-sourced LDM, such as Stable Diffusion Rombach et al. (2022), SDXL Podell et al. (2023), and DeepFloyd IF, and the absence of the regulation to the utility of these models. Before the society reaches a consensus on the definition of fair use of LDM, adversarial attacks on LDM should be one of the most important tools for ordinary people to protect themselves against unpredictable copyright and privacy risks induced by LDM.

## 2 BACKGROUND

### 2.1 LATENT DIFFUSION MODEL

LDM (Rombach et al., 2022) learns to generate the data in two stages.

**VAE Stage** VAE stage uses an encoder $\mathcal{E}$ and a decoder $\mathcal{D}$ to map the real data $x$ and the latent variable $z_0$. Both the encoder $\mathcal{E}$ and the decoder $\mathcal{D}$ are implemented as a conditional Gaussian distribution centered on the output of a neural network, which takes $x$ and $z_0$ as the input, respectively. They are trained in the style of VQVAE (Van Den Oord et al., 2017).

$$q(z_0|x) = \mathcal{N}(f_{\mathcal{E}}(x), \sigma_{\mathcal{E}})$$
$$q(x|z_0) = \mathcal{N}(f_{\mathcal{D}}(z_0), \sigma_{\mathcal{D}})$$
$$\tag{1}$$

Note that the variances of these two Gaussian distribution are extremely small. Hence, we omit the variance and consider the mapping deterministically.

**Diffusion Stage** Diffusion stage perturbs the latent variable $z_0$ with Gaussian noise step by step in the forward process $q(z_{1:T}|z_0)$, generating a series of latent variables $z_{1:T}$. This process finally maps $z_0$ to an approximate standard Gaussian noise $z_T$. A reverse process $p_\theta(z_{0:T}) = p(z_T)\prod_{t\le T} p_\theta(z_{t-1}|z_t)$ is built to predict $z_{t-1}$ from $z_t$. Here, the starting point $p(z_T)$ is a standard Gaussian distribution, matching $z_T$ in the forward process. $p_\theta(z_{t-1}|z_t)$ are parameterized conditional Gaussian distribution given by the following definition:

$$p_\theta(z_{t-1}|z_t) = \mathcal{N}(z_{t-1}; \frac{1}{\sqrt{\alpha_t}}(z_t(z_0,\epsilon) - \frac{\beta_t}{\sqrt{1-\overline{\alpha}_t}}\epsilon_\theta(z_t,t)), \sigma_t \boldsymbol{I}) \tag{2}$$

In the remainder, we follow Ho et al. (2020) to define constants $\alpha_t, \overline{\alpha}_t, \beta_t$. Intuitively, $\epsilon_\theta(z_t,t))$ is a noise predictor that extracts Gaussian noise from the perturbed latent variable $z_t$. LDM then exploits this predicted noise to denoise the $z_t$ and finally recover it to $z_0$. The LDM parameter $\theta$ is optimized by minimizing the lower bound of the $p_\theta(z_0)$, which is approximately simplified to the following training loss:

$$\mathcal{L}_{LDM} = \mathbb{E}_{z\sim q(z), \epsilon\sim\mathcal{N}(0,1), t}\|\epsilon_\theta(z_t,t) - \epsilon\|_2^2 \tag{3}$$

**LDM-driven Few-shot Generation** LDM shows amazing abilities in few-shot image generation. Few-shot generation samples images based on a few reference images. These images often share the same art style or demonstrate the same object. LDM-driven few-shot generation methods Meng et al. (2021); Gal et al. (2022); Ruiz et al. (2023); Roich et al. (2022); Zhang & Agrawala (2023) are able to sample images with similar art styles or contents with the reference image successfully. Most of these methods fine-tune LDM on the reference images with the loss in Equation 3 so that LDM learns their art styles or contents by learning to predict the noise on latents of these images. Therefore, the key to fail the LDM-driven few-shot generation is to fail the noise predictor $\epsilon_\theta(z_t,t))$ in predicting the noise accurately.

## 2.2 ADVERSARIAL ATTACKS ON LATENT DIFFUSION MODEL

Adversarial attacks on LDM generate adversarial examples by adding tiny perturbations to clean images. These adversarial examples resembles clean images visibly but cannot be used as reference images in LDM-driven few-shot generation, as demonstrated in Figure 2. Early works of this adversarial attack are specific for certain few-shot generation methods. For example, Photoguard (Salman et al., 2023) focuses on SDEdit (Meng et al., 2021) and AdvDM (Liang et al., 2023) targets for Textual Inversion (Gal et al., 2022). Generally, existing methods can be categorized into two groups.

**Attacking VAE** These attacks try to bias the latent variable $z_0$ of the image $x$. This is done by minimizing the distance between $z_0$ and the latent variable $z_0^\mathcal{T}$ of a target image $x^\mathcal{T}$. Since they only involve the encoder and the decoder, we denote them by *attacking VAE*.

$$\min_\delta D(z_0, z_0^\mathcal{T}), z_0 = f_\mathcal{E}(x+\delta), z_0^\mathcal{T} = f_\mathcal{E}(x^\mathcal{T}) \tag{4}$$

The main efforts of these attacks focus on the cherry-pick of the target image $x^\mathcal{T}$ and the distance metric $D(\cdot)$. Photoguard (Salman et al., 2023) picks the $l2$-norm as $D(\cdot)$. Mist (Liang & Wu, 2023) introduces a specific target image that improves the visualization performance. DUAW (Ye et al., 2023) finds that the SSIM between decoded latents is an expressive distance metric.

**Attacking UNet** UNet (Ronneberger et al., 2015) is adopted by LDM to instantiate the noise predictor $\epsilon_\theta(z_t,t)$. This group of adversarial attacks try to fail the noise predictor from accurately predicting the noise of adversarial latent variables. AdvDM (Liang et al., 2023) does this by maximizing the training loss of LDM in Eq 3. Anti-Dreambooth (Van Le et al., 2023) and UDP (Zhao et al., 2023) roughly follows the objective of AdvDM and introduces the poisoning setup to counter Dreambooth (Ruiz et al., 2023), a popular few-shot genaration methods. Empirically, attacking UNet appears to be more powerful than attacking VAE since it considers both UNet and VAE.

## 3 IMPROVING TARGETED ATTACK

### 3.1 TARGETED ATTACKS

**General Form of Targeted Objective Functions**   We focus on attacking UNet since it has stronger performance than attacking VAE empirically Salman et al. (2023); Liang et al. (2023). Existing adversarial attacks on UNet Liang et al. (2023); Van Le et al. (2023) maximizes the training loss of LDM. This objective is exactly to maximize the summation of KL-divergence between $q(z_{t-1}|z_t, z_0)$ and $p_\theta(z_{t-1}|z_t)$.

$$\max_\delta \mathcal{L}_{LDM} = \max_\delta \mathbb{E}_q \sum_{t>1} D_{KL}(q(z_{t-1}|z_t, z_0)||p_\theta(z_{t-1}|z_t))$$
$$= \max_\delta \mathbb{E}_{z \sim q(z), \epsilon \sim \mathcal{N}(0,1), t} \|\epsilon_\theta(z_t, t) - \epsilon\|_2^2 \tag{5}$$

Adversarial attacks can be intuitively explained as maximizing the distance between the model output and the ground truth by adding adversarial perturbations to the model input. The distance serves as the objective function. A common approach to improve the empirical performance of adversarial attacks is to replace the ground truth in the objective function with a target and minimize the new objective function (Liu et al., 2016; Carlini & Wagner, 2018; Dong et al., 2018; Qin et al., 2019). Intuitively, this alternative goal makes the adversarial example similar to the target from the perspective of the neural network and therefore confuses the network.

Inspired by this idea, the objective function in Equation 5 can be considered as the distance between the ground truth $q(z_{t-1}|z_t, z_0)$ and the model output $p_\theta(z_{t-1}|z_t)$. By replacing the ground truth $q(z_{t-1}|z_t, z_0)$ with a target, we can introduce an alternative targeted objective function for adversarial attack on LDM. Note that the ground truth here is a distribution rather than a fixed value. Hence, the target should be a distribution. We denote the target distribution as $\mathcal{T}_t$. The general form of targeted objection function for adversarial attack on LDM can be then formulated as follows:

$$\min_\delta J = \min_\delta \mathbb{E}_q \sum_{t>1} D_{KL}(\mathcal{T}_t||p_\theta(z_{t-1}|z_t)) \tag{6}$$

**Some Trivial Targeted Objective Functions**   A natural choice to factorize $\mathcal{T}_t$ is $q(z_{t-1}^{\mathcal{T}}|z_t^{\mathcal{T}}, z_0^{\mathcal{T}})$, because it has the same form with the ground truth distribution. The only difference is the condition latent variable $z_0^T = f_\varepsilon(x^{\mathcal{T}})$. $x^{\mathcal{T}}$ is a human-chosen target image other than the clean image to be attacked. This distribution is a determined Gaussian distribution conditioned on $z_t^{\mathcal{T}}$ Ho et al. (2020)

$$q(z_{t-1}^{\mathcal{T}}|z_t^{\mathcal{T}}, z_0^{\mathcal{T}}) = \mathcal{N}(z_{t-1}^{\mathcal{T}}; \frac{\sqrt{\overline{\alpha}_{t-1}}\beta_t}{1-\overline{\alpha}_t}z_0^{\mathcal{T}} + \frac{\sqrt{\alpha_t}(1-\overline{\alpha}_{t-1})}{1-\overline{\alpha}_t}z_t^{\mathcal{T}}, \frac{1-\overline{\alpha}_{t-1}}{1-\overline{\alpha}_t}\beta_t)$$
$$= \mathcal{N}(z_{t-1}^{\mathcal{T}}; \frac{1}{\sqrt{\alpha_t}}(z_t^{\mathcal{T}}(z_0^{\mathcal{T}}, \epsilon) - \frac{\beta_t}{\sqrt{1-\overline{\alpha}_t}}\epsilon), \frac{1-\overline{\alpha}_{t-1}}{1-\overline{\alpha}_t}\beta_t) \text{ (reparameterization)} \tag{7}$$

LDM does the parameterization in Equation 2. Combining this parameterization and Equation 7, the targeted objective function is then determined, where we unify the noise $\epsilon$ in the sampling of $z_t$ and $z_t^{\mathcal{T}}$, following the idea in Van Le et al. (2023).

$$\min_\delta J = \min_\delta \mathbb{E}_q \sum_{t>1} D_{KL}(q(z_{t-1}^{\mathcal{T}}|z_t^{\mathcal{T}}, z_0^{\mathcal{T}})||p_\theta(z_{t-1}|z_t))$$
$$= \min_\delta \mathbb{E}_{z_0^{\mathcal{T}}, z_0, \epsilon, t} \frac{1}{2\sigma_t^2 \sqrt{\alpha_t}} \|(z_t(z_0, \epsilon) - z_t^{\mathcal{T}}(z_0^{\mathcal{T}}, \epsilon)) - \frac{\beta_t}{\sqrt{1-\overline{\alpha}_t}}(\epsilon_\theta(z_t(z_0, \epsilon), t) - \epsilon)\|_2^2$$
$$= \min_\delta \mathbb{E}_{z_0^{\mathcal{T}}, z_0, \epsilon, t} \frac{1}{2\sigma_t^2} \|(z_0 - z_0^{\mathcal{T}}) - \frac{\beta_t}{\sqrt{\alpha_t}\sqrt{1-\overline{\alpha}_t}}(\epsilon_\theta(z_t(z_0, \epsilon), t) - \epsilon)\|_2^2$$
$$\tag{8}$$

Note that Van Le et al. (2023) also introduces a targeted objective function. This objective function differs from Equation 8 only by removing $z_0$.

Following Van Le et al. (2023), we choose a real image (a portrait) as the target image. However, the empirical result shows that both this targeted objective function and the one given by Van Le et al. (2023) fail in successfully attacking LLM-driven few-shot generation under perturbation constraint $4/255$. This is also cross-validated by the visualization result in Van Le et al. (2023) that its target objective is inferior to its untargeted objective. To further investigate the impact of introducing targets, we visualize the prediction error of U-Net and give explanation in Appendix B.2 and C

### 3.2 IMPROVING TARGETED ATTACKS

**Target Distribution $\mathcal{T}_t$**   We restart from the general form of the targeted objective function given by Equation 6. The first question is, what is the form of our target distribution $\mathcal{T}_t$?

We first determine a prior that we still confine the distribution to be Gaussian with a constant variance, because this makes the KL-divergence in Equation 6 tractable. As a result, we only need to determine the mean of the target distribution.

As our intention is to "trick" LDM by inducing it to predict the wrong distribution, the mean of our target distribution should be very different from that of the conditional distribution $p_\theta(z_{t-1}|z_t)$. Note that for a trained LDM, the mean of $p_\theta(z_{t-1}|z_t)$ must fit $\frac{1}{\sqrt{\alpha_t}}(z_t - \frac{\beta_t}{\sqrt{1-\overline{\alpha}_t}}\epsilon)$, where $\epsilon$ is the Gaussian noise that predicted by $\epsilon_\theta(z_t, t)$. An intuitive idea is that for any $\epsilon$, the prediction by $\epsilon_\theta(z_t, t)$ collapses to a fixed value. We give highly intuitive and visualized comparison between different choice of target distributions, showing that fixing a uniform target $T$ does trick Unet to predict worse.

However, not every fixed value works in attracting $\epsilon_\theta(z_t, t)$ under the scale constraint of adversarial perturbation $\delta$. We observe that this objective function introduces semantic of the target image $x^\mathcal{T}$ to the output image of SDEdit Meng et al. (2021), a few-shot generation method. The introduced semantic acts as a very strong visual distortion to the output image. Intuitively, we can also add the semantic of the target image into the output image of Text-to-Image process by enforcing the denoising procedure to predict a noise that is close enough to $z_0^\mathcal{T}$. Thus, we also set our target latent to be $z_0^\mathcal{T} = f_\varepsilon(x^\mathcal{T})$, where $x^\mathcal{T}$ is a target image that is very different from natural images. To conclude, we finally factorize the target distribution by $\mathcal{T}_t = \mathcal{N}(\frac{1}{\sqrt{\alpha_t}}(z_t - \frac{\beta_t}{\sqrt{1-\overline{\alpha}_t}}z_0^\mathcal{T}), \sigma_\mathcal{T})$. Here, $z_0^\mathcal{T} = f_\varepsilon(x^\mathcal{T})$ and $\sigma_\mathcal{T}$ is any constant. Our Improved Targeted Attack (ITA) is finally determined:

$$\min_\delta J = \mathbb{E}_{\boldsymbol{z_0}, \boldsymbol{\epsilon}, t} \|\epsilon_\theta(z_t(\boldsymbol{z_0}, \boldsymbol{\epsilon}), t) - z_0^T\|_2^2 \tag{9}$$

Additionally, the attack on the VAE Salman et al. (2023); Liang & Wu (2023) can be jointly optimized with our targeted objective function. Note that existing attacks on the VAE of LDM also exploit a target image and minimize the distance between the adversarial example and the target image in the latent space of the VAE. Let $D(\cdot)$ be the distance metric. We determine a joint targeted objective function as follows, denoted by ITA+.

$$\min_\delta J = \mathbb{E}_{\boldsymbol{z_0}, \boldsymbol{\epsilon}, t} \alpha \|\boldsymbol{z_0} - z_0^\mathcal{T}\|_2^2 + \|\epsilon_\theta(z_t(\boldsymbol{z_0}, \boldsymbol{\epsilon}), t) - z_0^\mathcal{T}\|_2^2 \tag{10}$$

**Target Image**   Target image $x^\mathcal{T}$ does great impact on the performance of the targeted objective function. For cherry-picking the target image $x^\mathcal{T}$, we consider two kinds of images: natural images and images with artificial patterns. The former is used in the targeted Anti-DB (denoted by Anti-DB-T for short) (Van Le et al., 2023), while the latter is used in Liang & Wu (2023) as the target image for VAE attack.



Figure 3: Target image.

We first conduct an experiment to determine which kinds of images we should use. We use these two kinds of images as the target image in both Anti-DB-T and ITA (Equation 9). The result visualized in Figure 4 shows that natural images lead to minor visual effects in the output images of

---

**Algorithm 1** Improved Targeted Attack (ITA)

---

1: **Input:** Image $x$, LDM $\theta$, learning rates $\alpha, \gamma$, epoch numbers $N, M, K$, budget $\zeta$, loss function $\mathcal{L}_{LDM}$ in Equation 3, objective function $J$ in Equation 9 & Equation 10.

2: **Output:** Adversarial example $x'$

3: Initialize $x' \leftarrow x$.

4: **for** $n$ from 1 to $N$ **do**

5:    **for** $m$ from 1 to $M$ **do**

6:       $\theta \leftarrow \theta - \gamma \nabla_\theta \mathcal{L}_{LDM}(x', \theta)$

7:    **end for**

8:    **for** $k$ from 1 to $K$ **do**

9:       $x' \leftarrow x' - \alpha \nabla_{x'} J$

10:      $x' \leftarrow \text{clip}(x', x - \zeta, x + \zeta)$

11:      $x' \leftarrow \text{clip}(x', 0, 255)$

12:    **end for**

13: **end for**

---

attacked few-shot generation methods. By contrast, artificial patterns leave obvious texture on the output images. We show the detailed discussion in Appendix B.1. As the result, we pick the target image from Liang & Wu (2023) (See in Figure 3) as our target image.

We furthermore investigate how two basic properties of the artificial patterns, impact the effectiveness of the targeted attack. Here, pattern repetition means the number that the basic pattern, the word *MIST*, repeats in the target image. We tune the contrast and pattern repetition of the image in Figure 3 and produce several different images (shown in Figure 5). We then use these images as the target images in ITA and compare the output images of LoRA under the attack of ITA. Details of this experiment are also given in Appendix B.1. The result is visualized in Figure 5, which indicates that both the contrast and the pattern repetition should not be too low.

To conclude, we recommend to use target images that consist of artifical patterns with **sharp edges, high contrast, and dense patterns**, as the target image shown in Figure 3.

**Poisoning Attack on LoRA** Existing research shows that LDM is vulnerable to poisoning attack Ye et al. (2023); Van Le et al. (2023). We factorize the objective function of the poisoning attack setup with the targeted objective function given by Equation 9. Concretely, we fine-tune LDM with the adversarial examples for one step and optimize the adversarial examples with the trained LDM for one step, alternatively. We use LoRA Hu et al. (2021) with Dreambooth Ruiz et al. (2023) to fine-tune LDM, since it is the most popular LDM-driven few-shot generation method and thus our main target. The algorithm of this **I**mproved **T**argeted **A**ttack (ITA) is given in Algorithm 1.

## 4   IMPROVING MEMORY EFFICIENCY

We also work on improving the memory efficiency. The memory cost of our attack consists of three occupies, which store model weights, the computational graph, and optimizer states, respectively. In the adversarial attack, we only optimize the inputs so the memory used to store optimizer states is small. We mainly consider to save memory by decreasing the memory to store model weights and computational graph. Following mechanisms are introduced to our implementation of our attack.

**xFormers** We leverage xFormers (Lefaudeux et al., 2022) to reduce the memory cost of storing the computational graph. xFormers is a toolbox that provides memory-efficient computation operators for training and inference of transformer-based modules. We use their attention operator in the computation of cross attention layers in UNet.

**Gradient Checkpointing** Chen et al. (2016) A common tool of memory-efficient training is Gradient Checkpointing. Gradient Checkpointing separates a neural network into blocks. In forward-propagation, it only stores the activation. The back-propagation is done block by block. For each block, it reconstructs the forward computational graph within the block with the stored activation. Then, it constructs the backward computational graph within the block and compute the gradient over the activation. This greatly reduces the GPU memory at the cost of computing time. To balance the memory and time cost, we only apply gradient checkpointing in the down-block, mid-block, and up-block of the UNet.

| METHOD | MEMORY/GB |
|---|---|
| PHOTOGUARD | 6.16 |
| PHOTOGUARD+ | 16.79 |
| ADVDM | 6.28 |
| ANTI-DB | 7.33 |
| ITA&ITA+ | **5.77** |

Table 1: GPU memory cost of our method and baselines.

We evaluate the GPU memory cost of existing adversarial attacks on LDM and that of our attack. The setup of these baseline attacks and ours are demonstrated in Section 5.1. Note that PhotoGuard+

refers to the diffusion attack in Salman et al. (2023)/ The result in Table 1 shows that the GPU memory cost of our attack outperforms all baseline attacks. One key point is that our GPU memory cost is lower than 6GB. This means that **our attack is able to run on most of the consumer-level GPUs**, which helps popularize the application of adversarial attacks on LDM as a practical tool.

One concern of our memory-efficient attack is about the time cost. We also evaluate the running time of attacking 20 images with our attack, which is 1739 seconds on one NVIDIA RTX 4090 GPU. This running time is acceptable for users who protect their personal images since the sum of these images may not be large.

## 5  EXPERIMENT

Adversarial attacks on LDM aim at preventing unauthorized few-shot generation with LDM. In this section, we evaluate our proposed method on two state-of-the-art few-shot generation pipelines.

**SDEdit** (Meng et al., 2021): An image-to-image pipeline that modifies the content of single image. SDEdit can be used for fake photo creation concerning data privacy.

**LoRA** (Hu et al., 2021; Ruiz et al., 2023): The state-of-the-art LDM-driven few-shot generation method that finetunes the LDM with low-ranked adapters on dozens input images. LoRA generates high-quality images with similar items or styles of the input images. LoRA is the main concern about unauthorized few-shot artwork copying.

### 5.1  EXPERIMENTAL SETUPS

**SDEdit & Metrics** SDEdit generates one output image conditioned on one input withholding the structural similarity between them. To evaluate the structural similarity, we choose Multi-Scale SSIM (MS-SSIM) (Wang et al., 2003). In addition, we also want to assess the semantic distortion of the adversarial attacks. Thus, we also adopt CLIP Image-to-Image Similarity (denoted as CLIP-SIM) as our metric. Successful SDEdit keeps both the structural and semantic similarity high between the input and output images. Therefore, a strong adversarial attack is expected to have both these metrics low. Implementation details of SDEdit and two metrics are given in Appendix A.

**LoRA & Metrics** LoRA generates output images with the same content or in the same style of input images. LoRA does not guarantee the structural similarity between inputs and outputs. Hence, we directly assess the image quality of outputs. Our experiment first finetunes LDM with LoRA and then use the finetuned LDM to generate output images. Finetuning is done on 20 input images with the same content or style. We then generate 100 output images with the finetuned LDM and assess their image quality with CLIP-IQA (Wang et al., 2023a). An image with high quality have a low score of CLIP-IQA. Therefore, a strong adversarial attack should yield a high CLIP-IQA score. The implementation details of LoRA and CLIP-IQA are discussed in Appendix A.

**Resolution** The standard resolution for SD1.x is 512, while the one for SD2.x is 768. For cross-model transferability experiments, we set the resolution of every model to 512, disregarding that the standard resolution of SD2.1 is 768. The reason for this uniform resolution is to avoid the resizing, which may introduce distortion to the attacks. However, as LoRA on SD2.1 naturally generate image of resolution 768, we still test LoRA performance on SD2.1 on resolution 768.

**Datasets & Backbone Model** The experiment is conducted on CelebA-HQ (Karras et al., 2017) and Wikiart (Saleh & Elgammal, 2015). For CelebA-HQ, we select 100 images and each of 20 photos comes from an identical person. For Wikiart, we select 100 paintings that each of 20 paintings come from the same artist. We use Stable Diffusion 1.5 as our backbone model, for it enjoys the most active community in few-shot generation among all LDM-based generative models. Additionally, we also investigate the performance of our attack with Stable Diffusion 1.4 and Stable Diffusion 2.1 as backbone models to validate its cross-model transferability.

**Baselines** We compare our attack with existing open-sourced adversarial attacks on LDM, including AdvDM (Liang et al., 2023), PhotoGuard (Salman et al., 2023), and Anti-Dreambooth (Anti-DB) (Van Le et al., 2023). The diffusion attack in Salman et al. (2023) is denoted by PhotoGuard+. The implementation of baselines is detailed in Appendix A.

| | CELEBA-HQ | | | WIKIART | | |
| --- | --- | --- | --- | --- | --- | --- |
| | SDEDIT | | LORA | SDEDIT | | LORA |
| | MS-SSIM ↓ | CLIP-SIM ↓ | CLIP-IQA ↑ | MS-SSIM ↓ | CLIP-SIM ↓ | CLIP-IQA ↑ |
| NO ATTACK | 0.88 | 93.38 | 20.66 | 0.62 | 89.77 | 22.88 |
| ADVDM | 0.81 | 83.41 | 24.53 | 0.30 | 85.29 | 34.03 |
| PHOTOGUARD | 0.86 | 89.24 | 27.52 | 0.62 | 88.01 | 37.52 |
| ANTI-DB | 0.82 | 84.12 | 33.62 | 0.30 | 87.25 | 46.74 |
| ITA | 0.73 | 74.70 | 31.46 | **0.23** | **76.13** | 40.54 |
| ITA+ | **0.69** | **67.47** | **35.68** | 0.29 | 76.07 | **48.53** |

Table 2: Comparision of baseline methods and ours over two few-shot generation pipelines

| VICTIM | SD1.4 | | | SD1.5 | | | SD2.1 | | |
| --- | --- | --- | --- | --- | --- | --- | --- | --- | --- |
| BACKBONE | SDEDIT | | LORA | SDEDIT | | LORA | SDEDIT | | LORA |
| | MS ↓ | CS ↓ | CI ↑ | MS ↓ | CS ↓ | CI ↑ | MS ↓ | CS ↓ | CI ↑ |
| NO ATTACK | 0.85 | 91.71 | 20.32 | 0.85 | 91.16 | 19.22 | 0.80 | 79.00 | 16.78 |
| SD1.4 | 0.73 | 77.24 | 38.13 | 0.73 | 77.58 | 35.98 | 0.62 | 60.82 | 35.45 |
| SD1.5 | 0.73 | 77.29 | 36.65 | 0.73 | 77.50 | 32.11 | 0.72 | 60.10 | 45.05 |
| SD2.1 | 0.72 | 76.20 | 46.08 | 0.62 | 76.80 | 39.08 | 0.60 | 59.12 | 43.89 |
| VICTIM | SD1.4 | | | SD1.5 | | | SD2.1 | | |
| BACKBONE | SDEDIT | | LORA | SDEDIT | | LORA | SDEDIT | | LORA |
| | MS ↓ | CS ↓ | CI ↑ | MS ↓ | CS ↓ | CI ↑ | MS ↓ | CS ↓ | CI ↑ |
| NO ATTACK | 0.85 | 91.71 | 20.32 | 0.85 | 91.16 | 19.22 | 0.80 | 79.00 | 16.78 |
| SD1.4 | 0.67 | 66.83 | 40.69 | 0.67 | 66.40 | 31.53 | 0.58 | 56.41 | 67.96 |
| SD1.5 | 0.67 | 66.58 | 41.16 | 0.67 | 66.13 | 36.05 | 0.58 | 57.17 | 68.50 |
| SD2.1 | 0.67 | 66.33 | 41.80 | 0.67 | 57.17 | 41.96 | 0.58 | 57.27 | 73.59 |

Table 3: Cross-model transferability of ITA(top) and ITA+(bottom). MS, CS, and CI are our three metrics, MS-SSIM, CLIP-SIM, and CLIP-IQA, for short, respectively.

**Hyperparameters & Implementation Details** We tune the iteration number $N$ to be 4. The budget $\zeta$ in all adversarial attacks is set as $4/255$, which is a tiny budget acceptable to artists. The finetuning is done for $M = 10$ iterations on the learning rate $\gamma = 1 \times 10^{-5}$ with LoRA in Alg 1. Loss weight $\eta$, Step length $\alpha$ and iteration number $K$ of the attack are set to be $1 \times 10^2$, $5 \times 10^{-3}$, and $50$, respectively. All experiments are done on an NVIDIA RTX 4090 GPU.

## 5.2 OVERALL RESULT

We compare our two proposed attacks, named ITA (Eq 9) and ITA+ (Eq 10), to existing adversarial attacks on LDM. The overall result of comparison in two LDM-driven few-shot generation methods are demonstrated in Table 2. Our attack outperforms all baseline methods in decreasing the quality of output images in few-shot generation methods and generalizes over both SDEdit and LoRA. We visualize the output image of few-shot generation under our attack in Figure 2 and Appendix D.

## 5.3 CROSS-MODEL TRANSFERABILITY

Adversarial attacks on LDM are white-box and model-dependent, because their objective functions are model-dependent. Hence, how these attacks have transferable performance across LDM-based text-guided image generative models other than the backbone model $\theta$ is crucial for their practicality.

We evaluate the cross-model transferability of our attacks on three LDM-based text-guided image generative models: Stable Diffusion 1.4, Stable Diffusion 1.5, and Stable Diffusion 2.1. Here, backbone means the model used in generating adversarial examples and victim means the model used in few-shot generation.

We use 100 images from CelebA-HQ, 20 in each of 5 groups, to generate adversarial examples on three backbone models and run SDEdit and LoRA with these adversarial examples on three victim models. Experimental setups stay the same with those stated in Section 5.1, except for adding the

| DEFENSE | NO DEFENSE | GAUSSIAN | | JPEG | | RESIZING | | SR |
|---|---|---|---|---|---|---|---|---|
| PARAMS | | $\sigma = 4$ | $\sigma = 8$ | $Q = 20$ | $Q = 70$ | 2x | 0.5x | |
| LoRA | 25.51 | 17.24 | 28.39 | 38.13 | 29.01 | 26.10 | 32.91 | 39.63 |
| SDEDIT | 0.54 | 0.70 | 0.64 | 0.75 | 0.75 | 0.79 | 0.76 | 0.45 |

Table 4: Performance under different anti-attack methods. The noise budget is set to $8/255$

strength of SDEdit to 0.4 to encourage the variety of the experiments. We post MS-SSIM (MS) and CLIP-SIM (CS) of the output images of SDEdit and CLIP-IQA of those of LoRA in Table 3. The result shows that adversarial examples generated by our attack can fail few-shot generation of LDM-based text-guided image generative models other than the backbone model. Our attack shows considerate cross-model transferability of our attack [1]. Visualizations are given in Appendix B.4

## 5.4 ITA VS DENOISING-BASED ADVERSARIAL DEFENSE

Denoising-based adversarial defense is the most straight-forward and threatening approach to counter adversarial attacks on LDM. These defense methods exploit hard-coded or model-based denoisers to purify the adversarial perturbations, by which they transfer adversarial examples back to clean images. These defenses can be used by malicious individuals to crack our protection to personal images and conduct unauthorized few-shot generation with them. Therefore, it is important to investigate how robust our attack is against denoising-based adversarial defenses.

We conduct an experiment on one 20-image group of the adversarial examples generated by ITA and ITA+ from CelebA-HQ. We use several denoising-based adversarial defense methods to denoise the adversarial perturbations on the example and then apply SDEdit and LoRA on these purified adversarial examples. The experimental setup follows the setup in Section 5.1 except for the adversarial perturbation constraint, which is set to be 8/255, twice of the original constraint. This is still a small constraint compared to the setup of existing adversarial attacks, such as 16/255 in PhotoGuard (Salman et al., 2023) and Anti-DB (Van Le et al., 2023). For the defense methods, we use Gaussian (Zantedeschi et al., 2017), JPEG (Das et al., 2018), Resizing (Xie et al., 2017), SR (Mustafa et al., 2019). For hyperparameters, Gaussian adds Gaussian noise of standard variance 4 and 8 to the adversarial example. We try two JPEG compression qualities, 20 and 70. For Resizing we have two setups, 2x up-scaling + recovering (denoted by 2x) and 0.5x down-scaling + recovering (denoted by 0.5x). All interpolation in resizing is bicubic.

The result is demonstrated in Table 4. We also visualize the output images in Appendix B.5. It shows that ITA and ITA+ still have strong impact on the output image quality of LDM-driven few-shot generation after processing by denoising-based adversarial defense. The CLIPIQA score of most cases even increase after the adversarial defense. A possible reason for the increase may be the image quality degradation introduced by the defense. The diffusion model is then affected by the degradation and produce lower quality outputs. The visualization in B.5 supports this point of view. Considering this effect, malicious infringers may not conduct multiple times of denoising. Hence, we believe that our adversarial attack has enough robustness to denoising-based adversarial defense. A noteworthy point is that ITA/ITA+ stay extra robust for resizing and high-quality JPEG compressing, which are two most common denoising methods. We believe the robustness enable ITA/ITA+ into real-world application of protecting copyrighted contents. For real-world usage, we also evaluate ITA under black-box settings. Details are given in Appendix B.3.

## 6 CONCLUSION

This paper proposes Improved Targeted Attack (ITA), a novel method of adversarial attack on LDM. ITA benefits from an improved targeted objective function and outperforms existing adversarial attacks in countering state-of-the-art LDM-based few-shot generation methods. Additionally, several mechanisms are introduced to decrease the GPU memory cost of ITA to lower than 6GB, making it available to run on most personal computers. ITA can serve as a practial tool for people exposed to the risk of copyright and privacy raised by LDM-based few-shot generation to protect themselves.

---

[1] We use 512 as the uniform resolution. The result of SD2.1 may be different under the resolution 768 (See Appendix B.4)

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

## A    IMPLEMENTATION AND EXPERIMENTAL DETAILS

**LoRA** For the evaluation, we finetune LDM with on CelebA-HQ dataset using the prompt *a photo of a sks person*, which was first used in the paper of Anti-Dreambooth (Van Le et al., 2023). This is because CelebA-HQ consists of portraits of certain people. Similarly, we use the prompt *a painting of sks style* on the WikiArt dataset, because WikiArt consists of paintings by certain artists. The number of finetuning epochs is set to be 1000 which ensures LoRA on clean images achieves considerate performance.

**SDEdit** The strength of SDEdit is set to be 0.3, which makes sense because a higher strength modifies the input image too much and a lower one keeps too many details of the input image. We use a null prompt to avoid the effect of prompts in the generation.

**Metrics** For MS-SSIM, we use the default setting in the implementation [2]. CLIP-SIM computes the cosine similarity between the input images and the output images in the semantic space of CLIP (Radford et al., 2021) and is given by the following definition:

$$\text{CLIP-SIM}(X, Y) = \cos(\mathcal{E}_{clip}(X), \mathcal{E}_{clip}(Y)) \qquad (11)$$

where $\mathcal{E}_{clip}$ is the vision encoder of the CLIP Radford et al. (2021) model. CLIP-IQA is a non-reference image quality assessment metric that computes the text-image similarity between the image and some positive & negative prompts. In the official implementation [3], the author exploits prompts such as *Good image*, *Bad image*, and *Sharp image*. An image with high quality is expected to have high text-image similarity with positive prompts and low text-image similarity with negative prompts. In our experiments, we use the positive prompt *A good photo of a sks person* and the negative prompt *A bad photo of a sks person* for CelebA-HQ and the positive prompt *A good photo of a sks painting* and the negative prompt *A bad photo of a sks painting* for WikiArt. Since we want to measure how poor the output image quality is, we use the text-image similarity between output images and the negative prompt. A strong adversarial attack results in low quality of output images and a high similarity between output images and the negative prompt.

**Baselines** We use the official implementation of Photoguard [4], AdvDM [5], and Anti-Dreambooth [6] in our experiments. For Photoguard, we follow the default setup in the official Python notebook file [7], except for tuning the adversarial budget to 4/255. For AdvDM, we also follow the default setup in the official implementation. For Anti-Dreambooth, we directly use the official implementation except for transferring it to LoRA. The default setup sets 10 steps of training LoRA and 10 steps of PGD attacks in every epoch. However, the default epochs of Anti-Dreambooth is too time-consuming. Therefore, we tune the total epochs of one single attack to be 4, which is a fair comparison for our method.

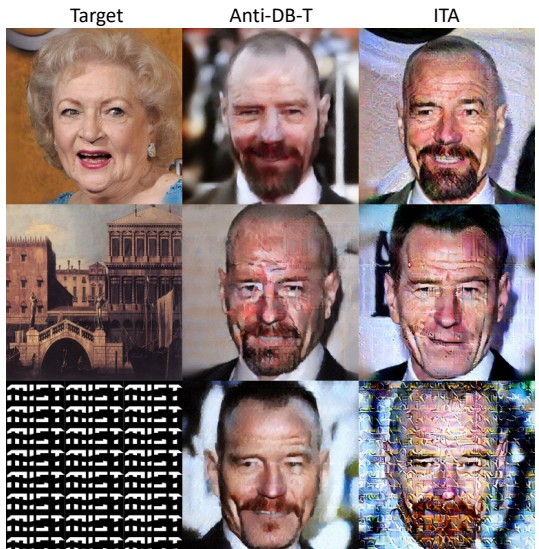

Figure 4: Comparison to a trivial targeted attack (Anti-DB-T) and trivial target images.

## B    ABLATION STUDIES

### B.1    DIFFERENT TARGET IMAGES

We conduct several experiments to validate our observation in how to select the target image $x^{\mathcal{T}}$ in Section 3.2.

---

[2]https://github.com/VainF/pytorch-msssim

[3]https://github.com/IceClear/CLIP-IQA

[4]https://github.com/MadryLab/photoguard

[5]https://github.com/mist-project/mist

[6]https://github.com/VinAIResearch/Anti-DreamBooth

[7]https://github.com/MadryLab/photoguard/

First, we pick two natural images, one from CelebA-HQ and the other from WikiArt, as our target image in ITA. Additionally, we try the Anti-DB-T Van Le et al. (2023) with these two target images to distinguish the difference between normal targeted attack and ITA. Following the setup in Section 5.1, we apply Anti-DB-T and ITA to LoRA. The output images of LoRA are visualized in Figure 4. As shown in the figure, only using ITA and a proper target image can achieve satisfying results. **This indicates that both our optimizing goal and our choice of target is the key to a successful attack**. What is also noteworthy is that when using natural images(person and paintings) as target, ITA still introduces the structural pattern in the output image, while Anti-DB-T shows none. This also illustrate the superiority of our methods.

Second, we investigate how the pattern of the target image impacts the attack performance. The result is visualized in Figure 5. When the pattern gets denser, the performance of the attack increases, but starts to decrease when the pattern is too dense. The increase is fairly understandable, as a denser pattern contains more semantic contents, which are introduced to the output images. We attribute the decrease to the modeling ability of diffusion model. That is, the diffusion model is not capable of modeling images of such intense semantic. Thus, the target latent cannot represents the semantic of the original target image. Therefore, the targeted method fails. As for the contrast, the performance of the attack increases as the pattern's contrast goes higher. The increase is under expectation as high contrast should give more semantic distortion.

Figure 5: Target images with different pattern repetition and contrast results in different effects.

### B.2 U-NET PREDICTION ERROR UNDER DIFFERENT TARGETS

As the final output of Text-to-Image process is a composition of several U-Net predictions, visualizing the U-Net prediction error should be an intuitive way to understand how different targets affects the Text-to-Image process. In general, we want the prediction error to be consistent under different timesteps. Thus the error can be accumulated through the denoising process and add distortion to the final output. We define U-Net prediction error as U-Net prediction difference between a clean sample and its corresponding adversarial one, under the same timestep $t$ and the same noise $\epsilon$. For targeted methods, we select our ITA method and targeted version of Anti-DB. The targeted image is Figure 3. We also select two un-targeted method: AdvDM and Anti-DB for comparison between targeted and un-targeted methods. We calculate the U-Net prediction error on a dataset of 20 image from CelebA-HQ.

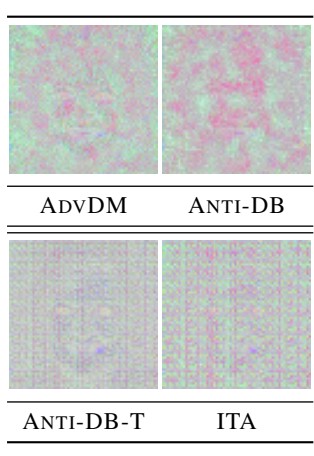

Table 5: Prediction error of U-Net under different attacks.

Intuitively, the difference of targeted Anti-DB and our ITA method would be more significant in larger timesteps, where the noise is taking the dominance. Thus, we randomly sample $t \in [0.9T, T)$, and repeat the sampling for 100 times for every image in the dataset. Then we take the mean of the U-Net prediction error as our final results. The results are shown in Table 5. From the results we can clearly see the difference between distinct methods. First, all prediction errors from targeted methods show a strong semantic meaning of the target image, while those from un-targeted methods show none, which may explain why targeted methods outperform un-targeted ones. Second, the errors from all other methods are visibly stronger than targeted Anti-DB(illustrated by the contrast and lightness of the visualization), which may be the reason of the inferiority of targeted Anti-DB.

### B.3 BLACK-BOX EVALUATION IN LoRA TRAINING

**Black-box Prompts** In real life usage, the attacker could have no knowledge about the prompt the malicious user is using for training LoRA. Thus in this section we give a black-box setting where the prompts in ITA and the prompts in actual training are different. We select a 20-image group from CelebA-HQ and use ITA to produce adversarial examples under noise budget $4/255$ and prompt "a photo of a sks person". We then train LoRA with four different prompts (the prompt for class data prior is changed correspondingly) that represents four levels of black-box settings. We list these four levels from white box to black box.

*Original*: "a photo of a sks person"

*Unknown keyword*: "a photo of a pkp person"

*Unknown keywords*: "a photo of a pkp woman"

*Unknown prompt*: "an image of a pkp woman"

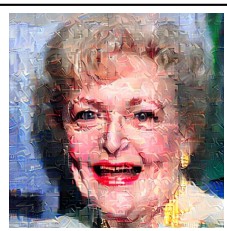

ORIGINAL          UNKNOWN KEYWORD

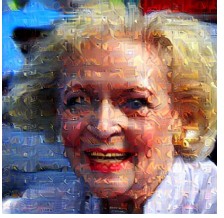
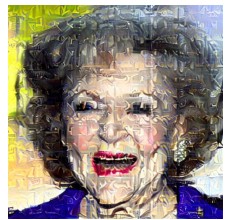

UNKNOWN KEYWORDS    UNKNOWN PROMPT

Table 6: LoRA output images with different prompts under the attack of ITA.

We then perform Text-to-Image on the finetuned model using the prompts separately. The experiment setup are still the same as Section 5.1. Visualization results are in Table 6. The result shows a degradation in the strength of attack when facing unknown keywords. However, when facing unknown class name(from "person" to "woman") or in unknown prompt settings, there is no further visible degradation. Meanwhile, there still exists strong visual distortion under all three scenarios.

**Black-box Case from Users** We also display a complete black-box case in Figure 6 from a community member, where we have no knowledge of the setup of LoRA training, including the model, the prompts and all other hyper-parameters. Our ITA method still owns a relatively strong visual result, which indicates ITA's capability for the real-world application.

### B.4 CROSS-MODEL TRANSFERABILITY

We visualize the output images of different victim models under ITA by different backbone models in Table 7. As shown in the table, our method shows a strong consistency among different models. An exception is that when using SD2.1 as the victim model, it tends to fail

Figure 6: Black-box case of ITA provided by an anonymous artist. LoRA is used for few-shot generation. The attack setup stays the same as the setup in Section 5.1.

in LoRA training(not learning the right person) instead of learning the strong semantic distortion from the target image. However, the model does learn the right person when no attack is conducted.

| VICTIM | SD1.4 | | SD1.5 | | SD2.1 | |
|---|---|---|---|---|---|---|
| BACKBONE | SDEDIT | LoRA | SDEDIT | LoRA | SDEDIT | LoRA |
| NO ATTACK | | | | | | |
| SD1.4 | | | | | | |
| SD1.5 | | | | | | |
| SD2.1 | | | | | | |

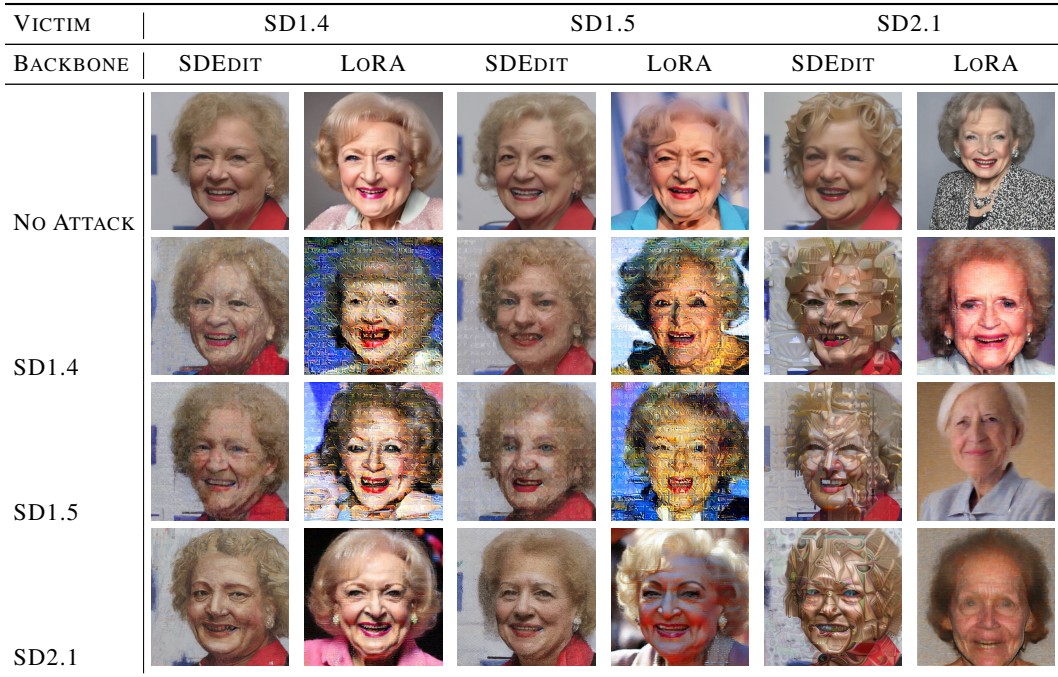

Table 7: Visualization of cross-model transferability of ITA.

| DEFENSE | NO DEFENSE | GAUSSIAN | | JPEG | | RESIZING | | SR |
|---|---|---|---|---|---|---|---|---|
| PARAMS | | $\sigma = 4$ | $\sigma = 8$ | $Q = 20$ | $Q = 70$ | 2x | 0.5x | |
| LoRA | | | | | | | | |
| SDEDIT | | | | | | | | |

Table 8: Performance under different anti-attack methods. The noise budget is set to $8/255$

Also, the SDEdit process is extra strong when victim is SD2.1. We attribute this phenomena to the resolution mismatch. SD2.1 is trained to receive images of resolution 768, while we actually fed it with images of resolution 512. This may leads to different behaviour of SD2.1. Also, the resolution mismatch between SD1.x and SD2.x may be the main reason for the performance degradation when using SD1.x as victim and SD2.1 as backbone.

## B.5 ITA VS DENOISING-BASED ADVERSARIAL DEFENSE

Table 8 visualizes the output images of SDEdit and LoRA referring to the adversarial examples which are processed by different denoising-based adversarial defense methods. For both method, they still have strong performance under most of the cases except for Gaussian Noise($\sigma = 8$) and JPEG compression (quality= 20). However, in the exception cases, the defense has added visible degradation to the image, which also heavily affect both LoRA and SDEdit process. For example, LoRA learns to produce images comprised of small squares due to a hard compression of quality 20. And SDEdit produces images of visible Gaussian noise when adding Gaussian noise of $\sigma = 8$ as defense. It's noteworthy that both ITA and ITA+ seems to be strengthened after SR defense, which is an intriguing phenomena.

## C   DISCUSSION: WHY IS TARGETED ATTACK BETTER?

Liang et al. (2023) yields theoretical proof that maximizing the training loss of LDM (Equation 5) leads to optimal attack that makes the adversarial examples out-of-distribution. However, untargeted attacks Liang et al. (2023); Van Le et al. (2023) that adopt this objective are inferior to our targeted attack in decreasing the output image quality of LDM-based few-shot generation (See visualization in Figure 2). One natural question is, why the optimal untargeted attack is beaten by targeted attacks?

This question may be answered by the **misalignment between human vision and generative models** in determining what the *worst* example should be like. Generative models are trained only to meet human's preference of good examples. An image with chaotic texture is undoubtedly a *bad* example for human vision. From Appendix B.2 where prediction error of U-Net is visualized, we can see the prediction error triggered by ANTI-DB is actually slightly stronger than ITA(illustrated by the lightness). However, the error caused by ITA shows more dense patterns (illustrated by the contrast) and achieve better result in human eyes. That is, ANTI-DB actually finds a better adversarial examples under the judgment from a diffusion model, while ITA finds a sub-optimal example. But in human (or other perception metrics) judgements, the sub-optimal example actually has worse visual results. This observation provides an intuitive explanation of the misalignment.

## D   VISUALIZATION

In this section, we visualize the comparison result between our proposed methods, ITA and ITA+, and baseline methods.

### D.1   ADVERSARIAL EXAMPLES

Figure 7 demonstrates adversarial examples generated by our adversarial attack, which resembles a real image since the adversarial budget $\zeta = 4/255$ is quite small.

### D.2   PERFORMANCE AGAINST SDEDIT

Figure 8 visualizes the output images of SDEdit under different adversarial attacks. All adversarial attacks are budgeted by $\zeta = 4/255$. Two proposed methods add obvious noise to the output image, compared to no attack and three baseline methods, Photoguard, AdvDM, and Anti-DB. Furthermore, we notice that ITA adds chaotic texture to the image and that ITA+ erases some contents of the image.

### D.3   PERFORMANCE AGAINST LORA

Figure 9, 10, 11, and 12 show the output images of LoRA under different adversarial attacks. All adversarial attacks are budgeted by $\zeta = 4/255$. Compared to the output image of no attack, three baseline methods add noise of different levels to the output image, among which Anti-DB is the strongest. This is natural because Anti-DB is specific for countering LDM-based few-shot generation methods. However, details of the training data can be still recovered by LoRA under the attack of baseline methods. By contrast, our two proposed methods outperform these baseline methods by destroying the details of the output image. ITA adds texture with sharp edges to the output image, making it completely unusable. ITA+ severely blurs and zigzags the output image. We also notice that the diversity of the output image decreases under the joint optimization attack.

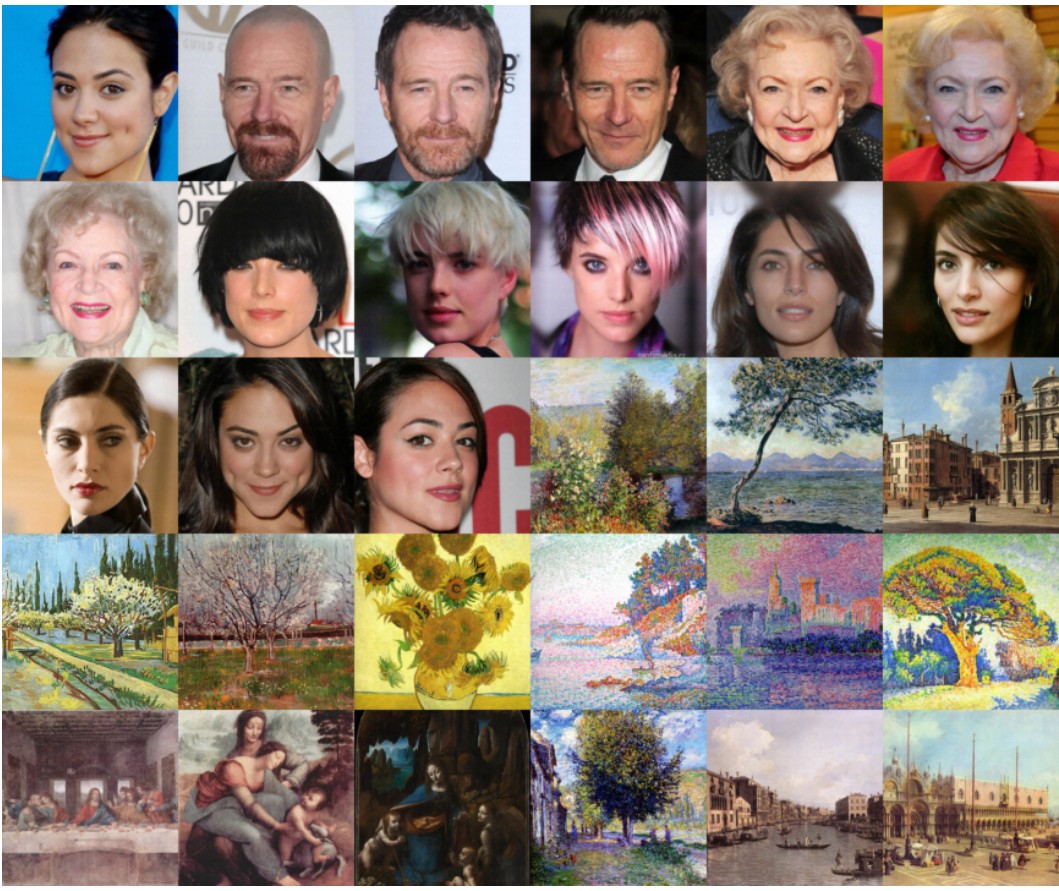

Figure 7: Adversarial examples on LDM generated by our proposed method with the adversarial perturbation budget $\zeta = 4/255$. The perturbation is quite small and almost human-invisible, making the adversarial examples resemble real examples.

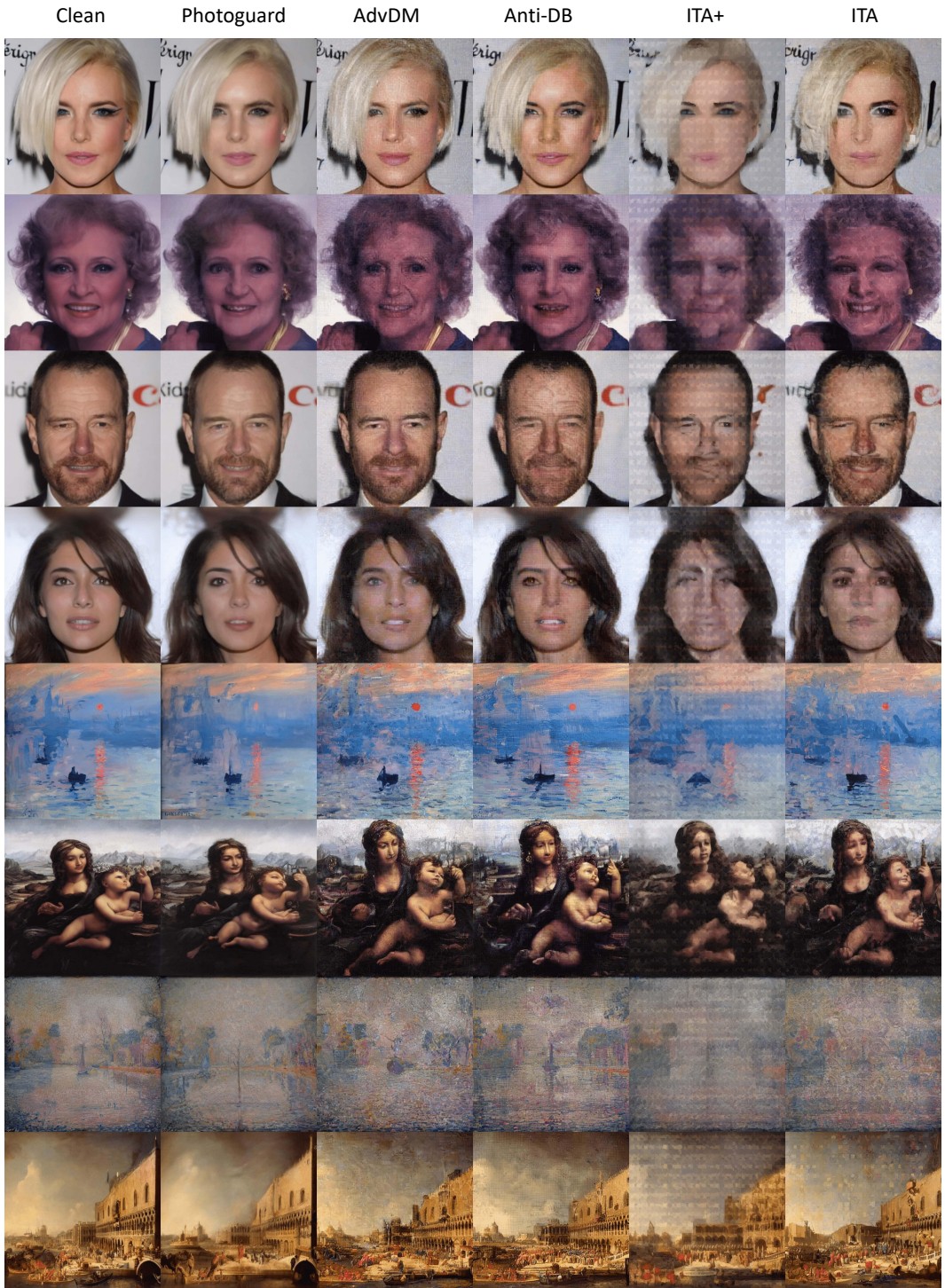

Figure 8: Output images of SDEdit under different adversarial attacks. With the same perturbation budget, our attacks better interfere the image quality compared to three baseline methods. Specifically, ITA adds chaotic texture to the image. ITA erases some contents of the image.

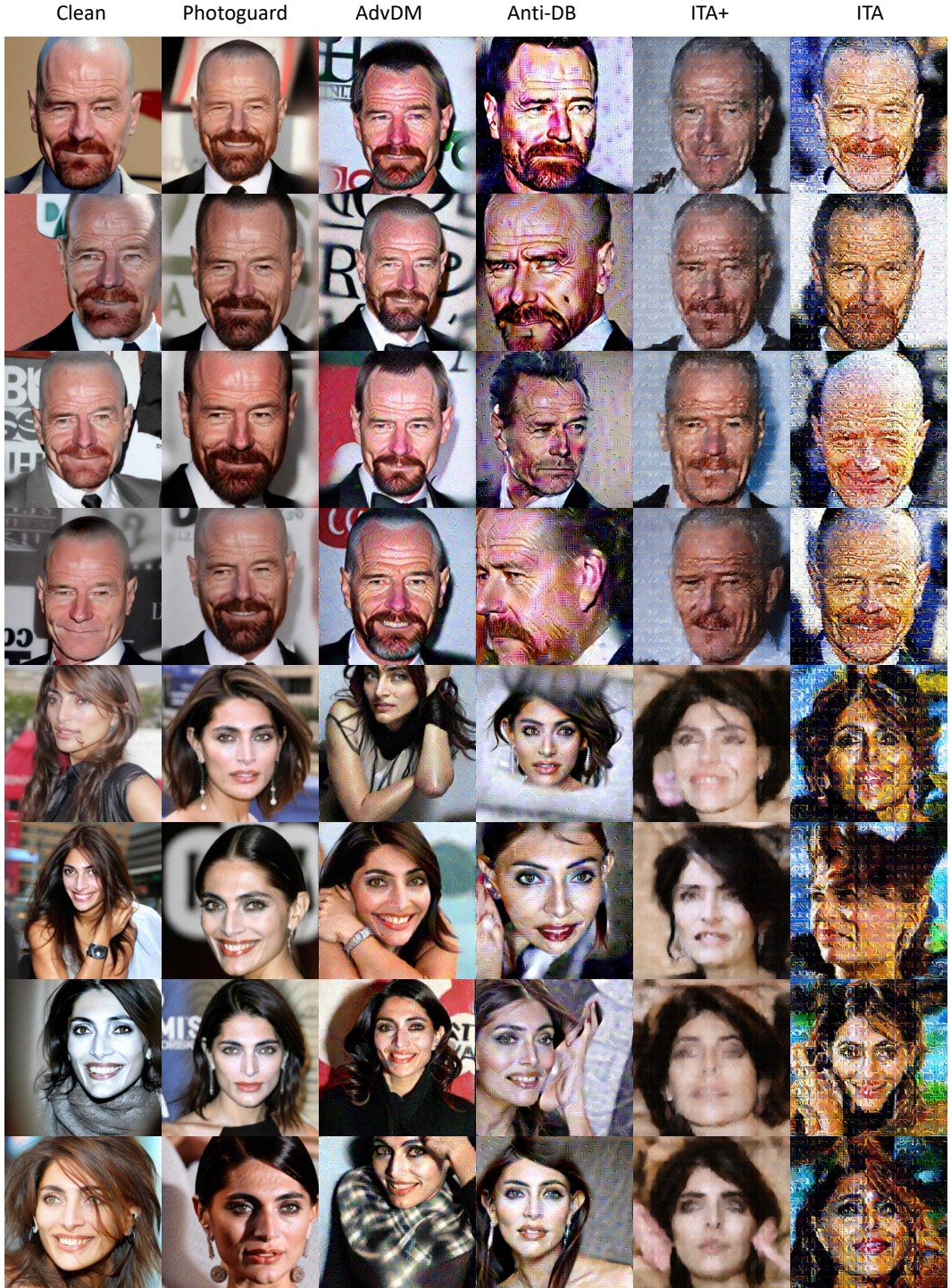

Figure 9: Output images of LoRA under different adversarial attacks. Two proposed methods outperform baseline methods. ITA adds texture with sharp edges to the output image. ITA+ severely blurs and zigzags the output image, decreasing the diversity of output images as well.

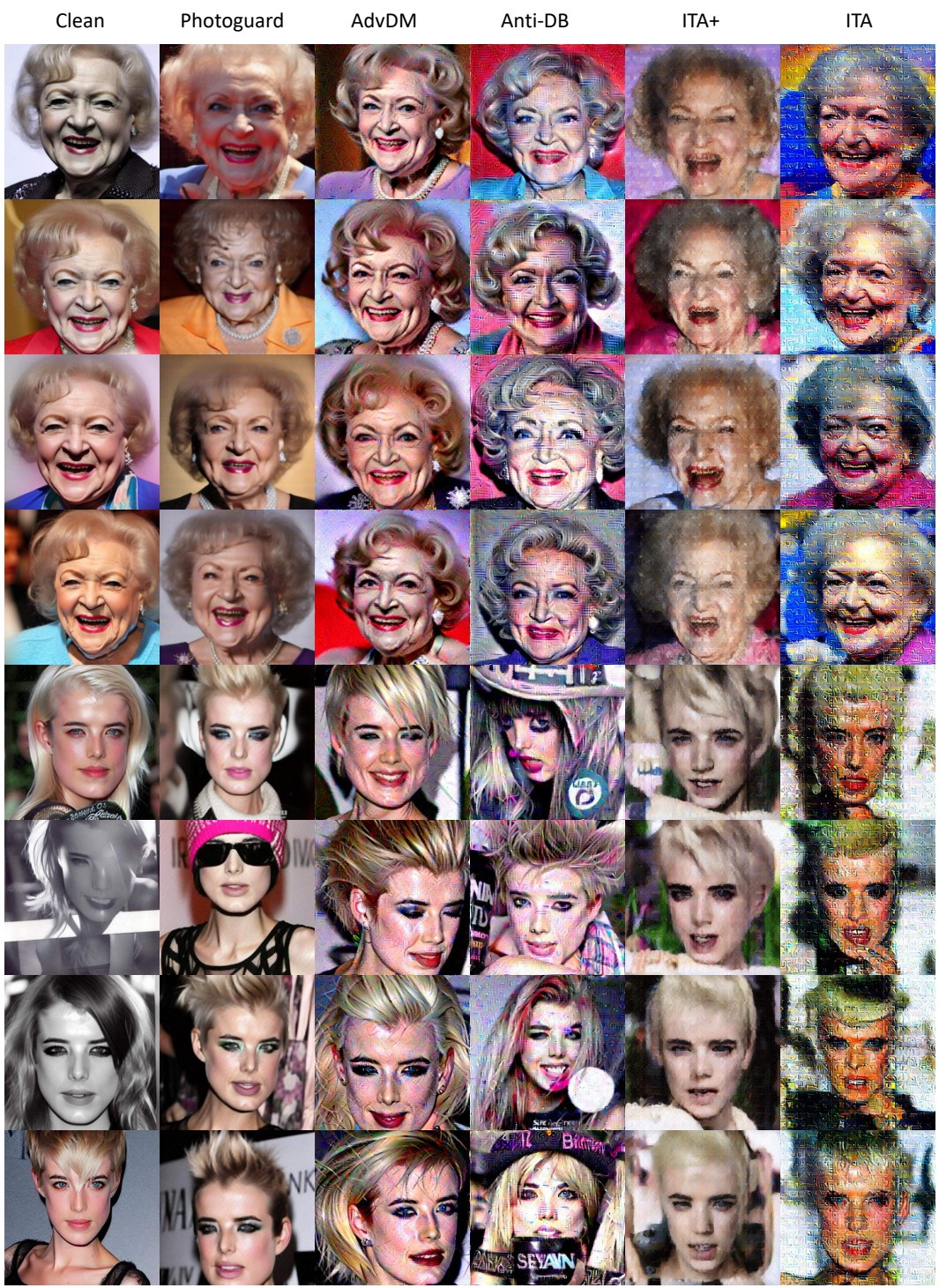

Figure 10: Output images of LoRA under different adversarial attacks. ITA and ITA+ outperform baseline methods. (Cond.)

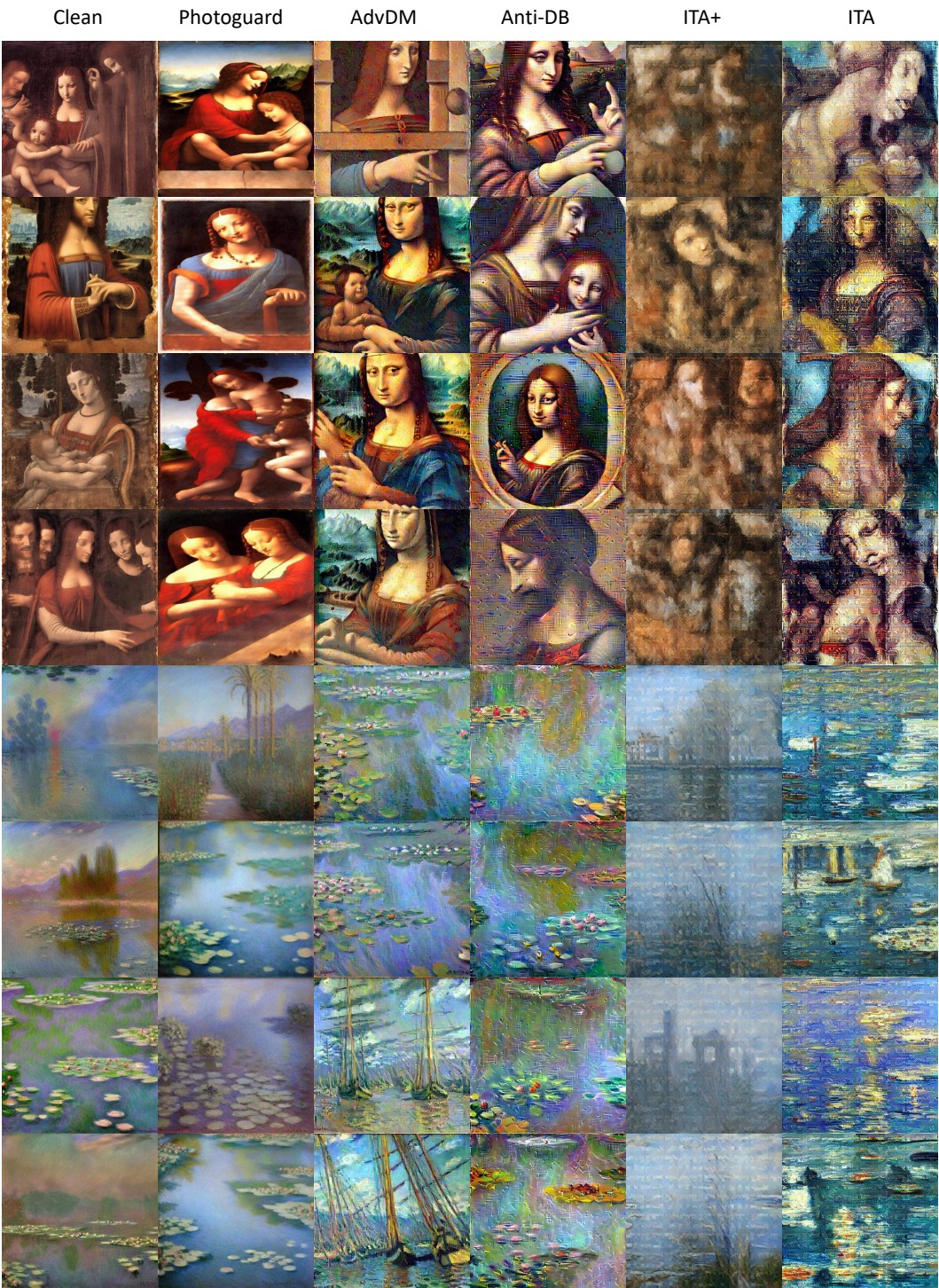

Figure 11: Output images of LoRA under different adversarial attacks. ITA and ITA+ outperform baseline methods. (Cond.)

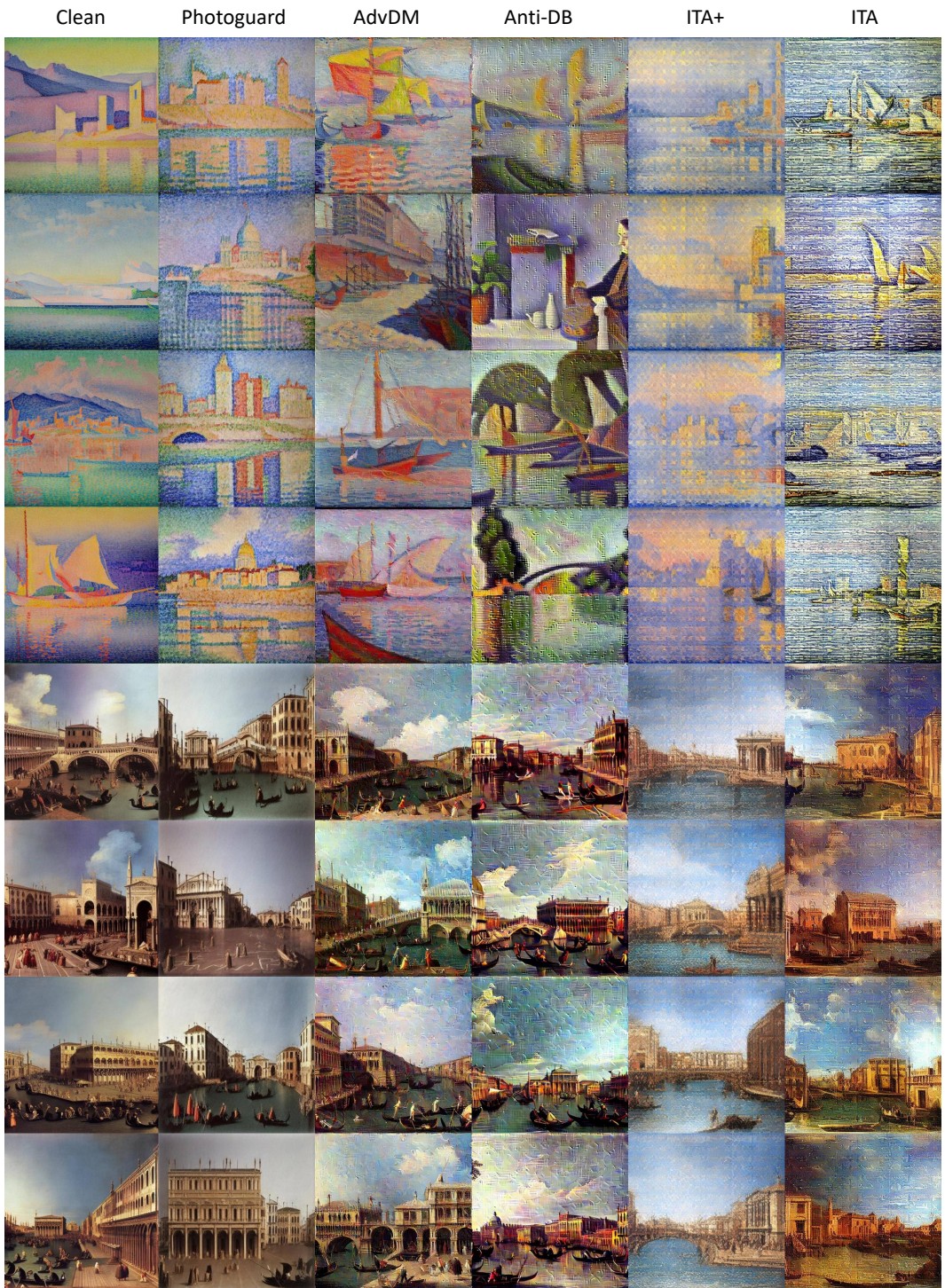

Figure 12: Output images of LoRA under different adversarial attacks. ITA and ITA+ outperform baseline methods. (Cond.)

