# OpenReview forum: "Understanding and Improving Adversarial Attacks on Latent Diffusion Model"
_ICLR.cc/2024/Conference — Submitted to ICLR 2024_

### Official Review · Reviewer_WV9g · 2023-10-23

**Soundness:** 3 good
**Presentation:** 3 good
**Contribution:** 3 good
**Rating:** 5
**Confidence:** 4

**Summary:**

This paper theoretically analyzes the adversarial attacks against latent diffusion models (LDM) for the mitigation of unauthorized usage of images. It considers three subgoals including the forward process, the reverse process, and the fine-tuning process. It proposes to use the same adversarial target for the forward and reverse processes to facilitate the attack. Experiments show it outperforms baselines and is robust against super-resolution-based defense.

**Strengths:**

1. This paper studies protecting images from being used by stable diffusion models without authorization, based on adversarial perturbations. This is an important research problem.

2. Different from existing empirical studies, this paper proposes a theoretical framework to help understand and improve adversarial attacks.

3. Experimental results show the proposed method can outperform existing baselines.

**Weaknesses:**

1. Not clear if this method needs white-box access to the subject LDM. That is, do the adversarial attackers use the same network used by infringers? Are the adversarial examples generated on one model/method still effective on different or unknown models/methods?

2. Only one defense method is evaluated. Are the adversarial samples robust to other transformations such as compression or adding Gaussian noises? Also, no adaptive defense is evaluated. If the infringers know about this adversarial attack, can they adaptively mitigate the adversarial effects?

3. From my understanding, Liang & Wu, 2023 and Salman et al., 2023 are not just "targeting the VAE" as claimed in this paper. They attacked the UNet as well.

4. Many issues in the writing.

    4.1. On Page 2, "serve as a means to" -> "mean".

    4.2. In Section 2.2, it says "As shown in Figure 1 adversarial attacks create an adversarial example that seems almost the same as real examples". However, Figure 1 only contains the generated images by the infringers instead of the adversarial examples as indicated by the title.

    4.3. The references to figures and tables are incorrect such as "Figure 5.3", "Table 5.2", "Table 5.3", etc. In the ablation study, the caption of the table is "Figure 5".

    4.4. Some math symbols are not defined where they first appear. For example, It would be better to mention that $\phi$ in Section 2.1 means the VAE. I suggest to use $\mathcal{E}\_{\phi}$ or $\mathcal{E}$ consistently. What does $\sqrt{\bar{\alpha\_t}}$ (the line below equation 5) mean? The text below Equation 10, for "$q\_{\phi}(v\_t | x')$ and $q\_{\phi}(v\_t | x')$", the second one should be $q\_{\phi}(v\_t | x)$. It would be better to briefly explain the N, M, and K in  Algorithm 2.

    4.5 The citation of SR in section E.2 is wrong. "Salman et al., 2023" -> "Mustafa et al. 2019".

**Questions:**

1. Could you explain why the equation 4 holds? Why do we need to use $q$ to express the left $p$?

2. Can the adversarial examples be effective for different or unknown models/methods?

3. According to section E.1, the target $\mathcal{T}$ in Equation 15 and 16 is defined as $\mathcal{E}(x^{\mathcal{T}})$. I can understand for the $\mathcal{L}\_{vae}^{\mathcal{T}}$, it's meaningful to encourage $\mathcal{E}(x')$ to be close to $\mathcal{E}(x^{\mathcal{T}})$. However, for the UNet part, what's the rationale to encourage the predicted noise at each timestep to be close to $\mathcal{E}(x^{\mathcal{T}})$? Because I think the final output $z_0$ should be close to $\mathcal{E}(x^{\mathcal{T}})$, but not the intermediate predicted Gaussian noise.

---

> ### Author Response · Authors · 2023-11-21
> **Response to the review**
>
> We thank the reviewer for the insightful review and feel sorry for the late response. We have done a major revision of our paper, removing the main part of our theory and complementing extensive experimental results. Please refer to our official comment.
>
> For the questions raised in the review, we will respond to them point by point.
>
>
> Weaknesses:
>
> 1.   Not necessarily. We are highly aware that in real world settings, the attacker can have no knowledge about the infringers(totally black-box). Thus, we conduct a cross-model transferability experiment for SD1.4/SD1.5/SD2.1(please refer to section 5.3). Results show that our method remains strong when facing unknown models. Moreover, we also test our method’s performance when the training prompt of LoRA is different from the one used for attacking. Our method also remains robust in this black-box setting(please refer to Appendix B.3). To simulate absolute black-box settings, we collect a piece of data from a community user (who generously shares his/her experiences), where we have no knowledge about any training detail, including the model and the prompts. The only thing we know is the noise budget is 4/255. In this case, our method still has satisfying distortion to the output image(please also refer to Appendix B.3).
>
> 2.   We have conducted experiments on other common defending methods (adding gaussian noises, JPEG compression and resizing). Numerical and visualization results show that our method is also robust to these defenses. We’ve also explored some adaptive defense strategies, and we highly agree on the academic significance of such defenses. However, as the goal of our method is to help the community to protect their copyrighted content, we think publishing techniques that aim to overcome the attack is not socially responsible. Therefore, we’ll not release any details of such strategies in our paper.
>
> 3.   Yes indeed. Thanks for your correction! We’ve corrected our description in the revision.
>
> 4.  Thanks for your detailed review, we’ve corrected and updated the paper based on the given points.
>
>       4.1  We’ve reconstructed our paper in a more fluent way and corrected grammar mistakes.
>
>       4.2  We’ve updated the references and the figures.
>
>       4.3  We’re sorry for the typographical errors. They have been fixed in the revision.
>
>       4.4  We did have inconsistent symbol definition in different sections. In the new version, we carefully choose our symbols and define them at the very beginning.
>
>       4.5  We’ve corrected our citation mistakes in the revision.
>
> Questions:
>
> 1.   We consider all latent variables in the LDM, including the latent variables z_{0:T} (reformulated as v_{0:T} in the first version of our paper) in the forward process and the latent variables z_{0:T} in the reverse process. We apply the sum operator over all these latent variables and yield Eq 4.
>
> 2.   Yes they can, as mentioned in the first point in the weaknesses section.
>
> 3.   The most intuitional way is indeed to let the output $z_0$ be close to $\mathcal{E}(x^T)$. And that is what the targeted version of Anti-DreamBooth intends to achieve(for detailed mathematical formulation of targeted Anti-DB please refer to section 3.1). However, results have shown the inferiority of this method, even compared to un-targeted ones. The intuition of our method is to “trick” the prediction of U-Net to have consistent errors under various timesteps, which can then be accumulated in the denoising process. By trying to “pull” the prediction close to a fixed target, we can control the prediction error to be consistent and have semantic similarity to the target ($\mathcal{E}(x^T)$), therefore, adding strong target-dependent semantic distortion to the output image. We have a great visualization for this intuition and give intuitive explainations for why targeted Anti-DB doesn’t work(Please refer to Appendix B.2).

---

> > ### Comment · Reviewer_WV9g · 2023-12-05
> >
> > Thank the authors for answering my questions. I think this is a very interesting paper. However, after checking the revised paper and other reviews, I agree that the changes in the rebuttal are significant and thus affect the original claims and contributions. So I lowered the score to 5.

---

### Official Review · Reviewer_19tS · 2023-10-30

**Soundness:** 3 good
**Presentation:** 3 good
**Contribution:** 3 good
**Rating:** 5
**Confidence:** 3

**Summary:**

The paper introduces a theoretical framework for understanding adversarial attacks on latent diffusion models. Based on this framework, the paper proposes a novel and efficient adversarial attack method that exploits a unified target to guide the attack process in both the forward and reverse passes of latent diffusion models.

**Strengths:**

1. The paper focuses on curbing the misuse of powerful LDM-based generative models for unauthorized real individual imagery editing, which is an important topic for securing privacy.

2. The theoretical foundation behind adversarial attacks on diffusion models is built, which contributes to the understanding of the behaviors of adversarial attacks.

**Weaknesses:**

1. More thorough examination accounting for a wider range of generative techniques could further validate the method's real-world utility and limitations.  While the proposed attack focuses on the prevalent LDM framework, its generalization to other powerful generative paradigms like SDXL, DALL-E, and Deep Floyd remains untested.


2. A more powerful baseline of PhotoGuard, i.e. Diffusion Attack is not compared to. This comparison could help gauge the true leadership of the new method. Without including this more powerful adversarial technique, the paper's claims about the proposed attack outperforming the current approaches remains uncertain.

3. The authors assert a memory-efficient design but do not provide details to support this claim. Further explanation or experimental evaluation of memory usage compared to alternative approaches would help validate the proposed method's efficiency advantages.

**Questions:**

Please refer to the weakness section.

---

> ### Author Response · Authors · 2023-11-21
> **Response to the review**
>
> We thank the reviewer for the insightful review and feel sorry for the late response. We have done a major revision of our paper, removing the main part of our theory and complementing extensive experimental results. Please refer to our official comment.
>
> For the questions raised in the review, we will respond to them point by point.
>
> Weaknesses:
> 1.  Unfortunately we don’t have enough time and resources to deploy SDXL and DeepFloyd IF and conduct experiments on them within the rebuttal period. But we evaluated our method on SD1.4,SD1.5 and SD2.1. Additionally, we add new experiments on cross-model transferability in Section 5.3. It shows that our attacks not only work well on Stable Diffusion 1.4 and Stable Diffusion 2.1 but also enjoy cross-model transferability that attacks whose backbone is SD1.5 work well on SD1.4 and SD2.1. Hence, we believe this can answer the question. Supported by cross-model experiment results, we’re quite confident about the performance of this method in the range of LDMs. We’ll continue on exploring our methods’ effect on more powerful generative paradigms in the future and update the results.
>
> 2.  We highly agree that the comparison between Diffusion Attack and ours is needed for deciding the leader of this field. However, the high demand of Diffusion Attack on GPU has been a challenge to our limited computation resources. Diffusion Attack requires a lot of time and more importantly, gpu memory. Under bf16/fp16 and our memory efficient optimization, Diffusion Attack still requires about 17G of gpu memory to run an attack on one single image and requires a much longer time for one round of PGD attack than other methods. This makes it hard to make a fair comparison between Diffusion Attack and ours. We’ll conduct the experiment as soon as our computation resource is upgraded(will soon!). Before we have the results, we add an additional explanation in the paper about this issue.
>
> 3.  Thank you for your advice! We admit the details in our original version are not sufficient, and we recognize the importance of memory efficiency, as most personal computers wouldn’t have as much GPU memory as researchers do. In further experiment, we find that the numerical results of original results is higher than the actual usage, possibly due to a more aggressive caching strategy on GPUs with larger memory. We retest our result after disentangling the caching factor and add more detailed information about our strategy. We’ve also tested other methods and made numerical evaluations, showing our advantages in memory efficiency. Please refer to section 4 for more details.

---

> > ### Comment · Reviewer_19tS · 2023-12-05
> >
> > After carefully reviewing the opinions of other reviewers and considering the revised version from the authors, I am inclined to lower my score to 5. I concur with reviewer FuAb's assertion that the impressive visualization results may be attributed to their selected target pattern. This is evidenced in Figure 5 in the Appendix, a detail I previously overlooked. Additionally, the original main contribution of the paper, "Our study provides the theoretical foundation for adversarial attacks on LDM," has been diminished, thereby weakening the strengths of the paper. Finally, the paper does not provide a comparison with the state-of-the-art method, Diffusion Attack, which leaves its contributions somewhat unclear. The authors are encouraged to further validate the technical merits through targeted revisions.

---

### Official Review · Reviewer_oH1F · 2023-10-31

**Soundness:** 1 poor
**Presentation:** 2 fair
**Contribution:** 1 poor
**Rating:** 3
**Confidence:** 4

**Summary:**

The paper proposed at a theoretical framework for adversarial attacks on Latent Diffusion Model (LDM).  The key to the theory is formulating an objective to minimize the expectation of the likelihoods conditioned on adversarial samples, of which the two terms implemented within the LDM explains the adversarial attack on the VAE and on the noise predictor, respectively. In addition, a new adversarial attack combining the two types of adversarial attacks are proposed.

**Strengths:**

1. Adversarial attacks on LDM is an interesting and practical problem.
2. Various experiments are conducted.

**Weaknesses:**

The proposed theoretical framework is not sufficiently innovative. In addition, the methodology exists many errors and the experimental verification are not sound. Specifically,

1. The key formulation of minimizing the conditional likelihood is trivial. The given theoretical proof is complicated. Actually, the likelihood equivalent to the KL divergence has been well-know. From this perspective, the proof is somehow trivial.

2. There exists many wrong equations.
a. In Eq. (5), the left term q(v_t|x) should be equal to the integral of the right term. Similar issue in Eq. (8).
b. In the first paragraph of Sect 3.2, q(v_t|v_{0:t-1} is mistakenly formulated.
c. In the last paragraph of Page 4, z_{t-1} is mistakenly formulated.
d. In Eq. (3), the sum in terms of z is mistakenly formulated given the expectation.
e. For Eq. (3), (4), (9)…, p()|x=x’ or p()|x’ is inappropriate, which should be put as the subscript or p(|x=x’).
f. The reformulation of z as v is unnecessary.

3. From the empirical results (Table 1), the strategy of combining two adversarial attacks does not perform significantly better than the Eq. (16). This raises doubt about the effectiveness of the newly proposed attack method.

Minor:
Some references are wrongly denoted, e.g. Figure 5.3.

**Questions:**

1. The proposed theoretical framework is not sufficiently innovative.

2. The methodology exists many errors

3. The experimental verification are not sound

---

> ### Author Response · Authors · 2023-11-21
> **Response to the review**
>
> We thank the reviewer for the insightful review and feel sorry for the late response. We have done a major revision of our paper, removing the main part of our theory and complementing extensive experimental results. Please refer to our official comment.
>
> As stated in our official comment visible to all reviewers, we remove the main part of the theory in the new version of our paper and replace it with the process of how we yield our methods intuitively. The main reason is that we find our theory cannot explain some observations, while the minor reason is the mistakes. We are sincerely sorry about that and really appreciate the insight review that drives us to rethink our original theory. We believe that replacing the theory with our intuition can clear most of the theoretical problems raised by the review.
>
> For the empirical results, as stated in our official comment, we conduct extensive experiments on cross-model validation, denoising countering, and black-box evaluation to validate the performance of our method. These results can prove that our method outperforms existing methods empirically and can serve as a practical tool.
>
> Although we remove the theory part in the new version of our paper, we will still respond to the theoretical questions raised by the review point by point based on the old version of our paper.
>
>
> Weaknesses:
>
> 1. We agree that the previous version of our paper includes theoretical results that are trivial and cannot explain some observations. Hence, we remove the theory and replace it with the intuitive process that yields the proposed method. We sincerely hope that the reviewer can read the new version of our paper and give advice, which will significantly help us refine our paper furthermore.
>
> 2. We will answer your question point by point
>
> a) Theoretically it should be the integral. However, empirically the variance of q(z_0|x) is extremely small (1e-6). Hence, q(z_0|x) is exactly a deterministic function that maps x to z_0. For this reason, we do not need to formulate q(z_t|x) as an integral.
>
> b) We agree that it is wrongly formulated. However, the theoretical speculation afterwards only exploits q(v_t|v_0), which has a closed form. Hence, this error does not influence the speculation a lot.
>
> c) We agree that it is wrongly formulated and should be $z_t\sim q(z_t|x)$.
>
> d) Eq 3. is the definition of the adversarial attack on LDM. Hence, we are not
> sure what mistakes there are in the definition. We guess it might be the formulation of expectation that we put the condition outside of the probability. We respond to this problem in e.
>
> e) p(), x=x’ is a mistaken expression. However, our expression is exactly $\mathbb{E}[p(|x)| x=x’]$. This expression is commonly used in statistics. Despite that, we correct this formulation according to the review in the new version of our paper.
>
> f) We agree that the reformulation is not necessary and cancel it in the new version.
>
> 3. We agree that the difference in performance cannot be explained by the theory in the first version of our paper. This is the main reason why we remove it in the new version of our paper. Instead, we attribute the success of Eq 16. (in the old version, now newly named Improved Targeted Attack, ITA) to the introduction of the target and the misalignment between human vision and diffusion models. Eq 15 (in the old version) now only serves as an easy-to-implement variant of Eq. 15 which sometimes shows superiority over SDEdit.
>
> Minor: Thanks for your correction, we’ve updated to the correct references.
>
> Questions:
>
> 1. We agree that the theoretical framework cannot work in explaining the result of our proposed methods. Hence, we replace it with the intuitive process about how we yield our proposed methods and try to provide some empirical observations on why our methods outperform. Again, we apologize for our mistaken introduction of the theoretical part in the first version of our paper and sincerely hope that the reviewer can give us more advice on the updated version.
>
> 2. See 1.
>
> 3. We highly understand the concern about our experimental verification. We’ve conducted more experiments including cross-model transferability, more ablation studies,denoising countering and so on. We’ve also added more visualization to side-prove our views and numerical results and added more details in experiment setups. We hope the new version of our paper can somewhat relieve the concern.
>
> Again, we apologize for the mistakes we have made in the first version of our paper. We try our best to correct and make up for them in the major revision. We will sincerely appreciate it if the reviewer reads the new version of our paper and gives further advice.

---

### Official Review · Reviewer_8G7v · 2023-11-01

**Soundness:** 3 good
**Presentation:** 3 good
**Contribution:** 3 good
**Rating:** 5
**Confidence:** 4

**Summary:**

This work proposes a method for improving adversarial attacks on Latent Diffusion Models (LDMs). The purpose is to generate adversarial examples preventing the LDMs from generating high-quality copies of the original images in order to protect the integrity of digital content. The authors mathematically introduce a theoretical framework formulating three sub-goals that existing adversarial attacks aim to achieve. This framework exploits a unified target to guide the adversarial attack both in the forward and the reverse process of LDM. The authors implement an attack method jointly optimizing three sub-goals, demonstrating that their method outperforms current attacks generally. The experiments are focused on the attacks on training pipelines, including SDEdit and LoRA.

**Strengths:**

1.	The author conducted extensive mathematical derivations, providing mathematical explanations for the existing methods.

2.	The experiments show that this method outperforms the baseline in most criteria and succeeded in attacking LoRA and SDEdit with Stable Diffusion v1.5.

**Weaknesses:**

1.	The only backbone model used in the experiments is Stable Diffusion v1.5.
There are plenty of more recent LDMs, such as Stable Diffusion v2.1 [1] and DeepFloyd/IF [2]. Will this method perform well in more advanced LDMs?

2.	The pseudo-code of algorithm 1 seems redundant and demonstrates nothing.
It literally equals to its description: “To optimize J_{ft}, we first finetune the model on x and obtain the finetuned model θ(x). Then, we minimize J_{ft} over x′ on the finetuned model θ(x).”


3.	The target image adopted by Liang & Wu (2023), visualized in “Figure E.1” (Is it mislabeled in Figure 8?), is the only target image used in the experiments. Did the authors try using different target images? Will the target image affect the effectiveness of this method?

4.	There are issues in the document layout. The labels of figures are mismatched to those in the texts.

5.	The offset problem needs to be clarified.
The authors claim that “The result in Figure 5.3 implies that offset takes place in 30% - 55% of pixel pairs in Δ_z_t and Δ_ε_θ, which means that maximizing J_q pulls Δ_z_t to a different direction of Δ_ε_θ and interferes the maximization of J_p.” Could the authors further explain it and Figure 3 in Section 4.1?

[1] Robin Rombach, Andreas Blattmann, Dominik Lorenz, Patrick Esser, and Björn Ommer. Highresolution image synthesis with latent diffusion models. In Proceedings of the IEEE/CVF Conference on Computer Vision and Pattern Recognition, pp. 10684–10695, 2022.

[2] Mikhail Konstantinov, Alex Shonenkov, Daria Bakshandaeva, and Ksenia Ivanova. Deepfloyd. https://github.com/deep-floyd/IF, 2023.

**Questions:**

1.	Will this method perform well in more advanced LDMs like Stable Diffusion v2.1 and DeepFloyd/IF?

2.	Will the target image affect the effectiveness of this method? Would the authors use other images as targets and test the effectiveness?

3.	Is that a typo in “In this tractable form, z′_t and z_t sampled from q_φ(v_t|x′) and q_φ(v_t|x′), respectively” below Equation 10?


4.	How does maximizing J_q pull Δ_z_t to a different direction of Δ_ε_θ and interfere the maximization of J_p? Could the authors further explain Figure 3 in Section 4.1?

---

> ### Author Response · Authors · 2023-11-21
> **Response to the review**
>
> We thank the reviewer for the insightful review and feel sorry for the late response. We have done a major revision of our paper, removing the main part of our theory and complementing extensive experimental results. Please refer to our official comment.
>
> For the questions raised in the review, we will respond to them point by point.
>
> Weaknesses:
>
> 1.   We add new experiments on cross-model transferability in Section 5.3. It shows that our attacks not only work well on Stable Diffusion 1.4 and Stable Diffusion 2.1 but also enjoy cross-model transferability that attacks whose backbone is SD1.5 work well on SD1.4 and SD2.1. Hence, we believe this can answer the question. As for DeepFloyd/IF, unfortunately we don’t have enough time and resources to deploy it and conduct experiments on it within the rebuttal. However, we’re quite confident about the performance of this method in LDMs, supported by cross-model experiment results. We’ll continue on supporting more LDMs in the future and update the results.
>
> 2.  We thank the reviewer for the advice. We have merged Algorithm 1 to original Algorithm 2 to become the new Algorithm 1 in the new version of our paper.
>
> 3.  Yes, it does. Our new experiments in Appendix B.1 show that picking natural images as the target image yields trivial performance. Section 3.2 also demonstrates new experiments to show that it would be better to pick images with tight patterns and high contrast as the target image.
>
> 4.  We have fixed this problem in the new version of our paper.
>
> 5.  We have removed the offset rate in the new version of our paper since we find that it cannot explain empirical observations.
>
> Questions:
>
> Experiments in Section 5.3 show that it performs well in SD2.1 even with backbone models of SD1.5. Due to the time limit, we do not evaluate it on DeepFloyd IF and SDXL. However, since it shares a similar structure (especially the VAE) and dataset with SD, we believe our attacks should also work well on it.
>
> 1.   Answered in the response of Weakness 2.
>
> 2.  No, the old version of our paper reformulates the intermediate variable z_t as v_t in the forward process. However, all these parts are removed in the new version.
>
> 3.  We remove this theory since we find that some observations cannot be explained by the theory.

---

> > ### Comment · Reviewer_8G7v · 2023-11-23
> >
> > Thanks for your response. In addition to the SD family, I think conducting evaluations on more recent LDMs could help to further validate the effectiveness of the proposed method.

---

### Official Review · Reviewer_FuAb · 2023-11-01

**Soundness:** 2 fair
**Presentation:** 2 fair
**Contribution:** 2 fair
**Rating:** 3
**Confidence:** 3

**Summary:**

Since diffusion and other advanced generative models have been used to replicate artworks and create fake content, a line work proposes a defense mechanism that adds a kind of adversarial perturbation to the protected images to prevent the adversary from fine-tuning their model on the images. This work proposes a more thorough theoretical formulation of the problem compared to the prior work and relies on this formulation to build an empirically stronger attack.

**Strengths:**

### Significance

I believe that this paper addresses an important problem with a widespread impact on both the technical community as well as society at large. The empirical results show a convincing improvement over the prior works on two different fine-tuning methods and two datasets.

I believe that introducing the target image for the adversarial objective and hence conducting a targeted attack instead of an untargeted one have a large effect on the empirical success of the attack.

**Weaknesses:**

### Correctness and clarity of the theoretical results

The paper formulates an adversarial optimization problem particularly tailored for the latent diffusion models (LDM). The analysis guides the algorithm design to some degree (more on this later). However, due to the lack of clarity and various approximations being introduced without proper justification, the theoretical results become less convincing. I will compile all my questions and concerns from Section 3 and 4 in place:

1. I am not sure what the sum $\sum_z$ is over in Eq. (3). The expectation is already over $z$ so I am a bit confused about the summation. My guess is that the sum is over all the latent variables in the diffusion process (different $z$’s in different steps). Is this correct?
2. If my previous understanding is correct, my next question is why should the adversary care about the latent variables in the intermediate steps of the diffusion process instead of, say, the final step of the inverse process before the decoder?
3. Based on the text, Eq. (3) should be equivalent to $\mathbb E_{z \sim p_{\theta}(z|x)}[- \log p_\theta(z|x')]$. My question is that a slightly different formula $\mathbb E_{z \sim p_{\theta}(z|x')}[- \log p_\theta(z|x) + \log p_{\theta}(z|x')]$ also seems appropriate (swapping order in the KL-divergence). Why should we prefer one to the other?
4. Section 3.2 uses the notation $\mathcal N(\mathcal E(x), \sigma_\phi)$ instead of $\mathcal N(f_{\mathcal E}(x), \sigma_{\mathcal E})$ from Section 2.1. Do they refer to the same quantity?
5. In the last paragraph of page 4, the Monte Carlo method must be used to estimate the mean of $p_\theta(z_{t-1}|x)$, but I cannot find where the mean is actually used. It does not seem to appear in Eq. (10) or in Appendix A.1. I also have the same question for the variance of $p_\theta(z_{t-1}|x)$ mentioned in the first paragraph of page 5.
6. Related to the previous question, it is mentioned that “the variance of $z_{t-1}$ is estimated by sampling and optimizing over multiple $z_{t-1}$.” It is very unclear what “sampling” and “optimizing” refer to here.
7. I do not quite see the purpose of Proposition 1. It acts as either a definition or an assumption to me. The last sentence “one can sample $x \sim w(x)$ from $p_{\theta(x)}(x)$” is also very unclear. Is the assumption that the true distribution is exactly the same as the distribution of outputs of the fine-tuned LDM?
8. $x^{(eval)}$ is mentioned in Section 3.4 but was never defined.
9. In Eq. (11), should both of the $\theta(x)$’s be $\theta(x')$ instead? Otherwise, $x'$ has no effect on the fine-tuning process of the LDM.
10. Section 4.1 is very convoluted (see details below).

### Issues with the offset problem and Section 4.1

**Comment #1**: I do not largely understand the purpose of the “offset” problem in Section 4.1. In my understanding, most of the discussion around the offset can be concluded by simply expanding the second term on the first line of Eq. (13):

$$
\sum_{t \ge 1}\mathbb E_{z_t,z'_t} || \Delta z_t +  \frac{\beta_t}{\sqrt{1 - \bar{\alpha_t}}} \Delta \epsilon ||_2^2
$$

$$
= \sum_{t\ge 1}\mathbb E_{z_t,z'_t} ||\Delta z_t||_2^2 + || \frac{\beta_t}{\sqrt{1 - \bar{\alpha_t}}}\Delta\epsilon ||_2^2 + \frac{2\beta_t}{\sqrt{1 - \bar{\alpha_t}}}\Delta z_t^\top\Delta\epsilon
$$

So the problem that prevents optimizing just the norm of $\Delta z_t$ and the norm of $\Delta \epsilon_\theta$ directly is the last term in the equation above (the dot product or the cosine similarity). I might be missing something here so please correct me if I’m wrong.

**Comment #2**: It is also unclear to me how the last line of Eq. (13) is reached and what approximation is used.

**Comment #3**: In theory, there is nothing preventing one from optimizing Eq. (13) as is. The issue seems to be empirical, but I cannot find the empirical results showing the failure of optimizing Eq. (13) directly and not using the target trick.

**Comment #4**: The authors “let *offset rate* be the ratio of pixels where the vector $\Delta z_t$ and $\Delta \epsilon_\theta$ have different signs.” If my understanding of the cosine similarity above is correct, this seems unnecessary and imprecise given that the cosine similarity is the exact way to quantify this.

**Comment #5**: In the first paragraph of page 7, it is mentioned that “meanwhile, since the original goal is to maximize the mode of the vector sum of…” I think instead of “mode,” it should be “magnitude” or the Euclidean norm?

### Empirical contribution

1. After inspecting the generated samples in Figure 11-15, my hypothesis is that the major factor contributing to the empirical result is the target pattern and the usage of the targeted attack. The pattern is clearly visible on the generated images when this defense is used, and this pattern hurts the similarity scores. This raises the question of whether the contribution comes from the theoretical formulation and optimization of the three objectives or the target. I would like to see an ablation study on this finding: (1) the proposed optimization + untargeted and (2) the prior attacks + targeted.
2. The choice of the target $\mathcal T$ is ambiguous. While the target pattern is shown in the Appendix, there is no justification for why such a pattern is picked over others and whether other patterns have been experimented with.

Overall, I believe that the paper can have a great empirical contribution, but it seems to be clouded by the theoretical analysis which appears much weaker to me.

**Questions:**

1. What are the approximations made on the fourth line of Eq. (22) and in Eq. (23)?
2. Why are MS-SSIM and CLIP-SIM used as metrics for SDEdit whereas CLIP-IQA score is used for LoRA? The authors allude to this briefly, but it still largely remains unclear to me.
3. The similarity metrics used in experiments seem to focus on the low-level textured detail rather than the style (please correct if this is not accurate). I am wondering if a better metric is the one that measures “style similarity” between the trained and the generated images. This might align better with the artwork and the copyright examples.
4. For the results reported in Table 1, how many samples or runs are they averaged over? Based on the experiment setup, 100 images are used for training the model in total for each dataset, and they are grouped in a subset of 20. So my understanding is that there are 100/20 = 5 runs where 100 images are generated in each run? Is this correct?
5. The fine-tuning hyperparameters for LoRA are mentioned in Section 5.1. Does the LoRA fine-tuning during the attack and the testing share the same hyperparameters? What happens when they are different (e.g., the adversary spends iterations during fine-tuning, etc.)? Can the proposed protection generalize?
6. Have the authors considered any “adaptive attack” where the adversary tries to remove the injected perturbation on the images (e.g., denoising, potentially via another diffusion model) before using them for fine-tuning?

---

> ### Author Response · Authors · 2023-11-21
> **Response to the review**
>
> We sincerely thank the reviewer for the insightful and detailed review, which greatly helps us in the improvement of the new version of our paper. As stated in our official comment visible to all reviewers, we remove the main part of the theory in the new version of our paper and replace it with the process of how we yield our methods intuitively. We believe this can solve most of the theoretical problems raised by the review. Also, we conduct novel ablation studies on target selection to answer the question about the empirical effectiveness of our method.
>
> Following are our point-by-point answers of the questions raised in the review. Although most of the theoretical problems are cleared by removing the old theory from the paper, we still respond to them based on the old version of our paper. We hope the reviewer can give opinions about the new version of our paper, which would be very valuable to us.
>
> Correctness and clarity of the theoretical results
>
> 1. Yes, it is the sum over all latent variables.
>
> 2. This is because the training and the sampling process is separate. Existing adversarial attacks usually attack the training process because attacking sampling process is very memory-expensive (imaging doing backprop throughout UNet for 25 or 50 times back to the input). Under this condition, considering the fact that the exact goal is to fail the sampling process and that the sampling process is done by step-by-step sampling of intermediate latent variables, it will be better if the attacker tries to make every intermediate latent variable out-of-distribution. This will interfere with every step of the sampling rather than just the final step.
>
> 3. This is an interesting idea. But as our original theory cannot well explain some observations, we would leave the discussion in the future work.
>
> 4. Yes, this is a clerical error.
>
> 5. We make a clerical error here that the proof should be given in Appendix A.2 of our first version, where $J_p$ is $J_{unet}$ in the Appendix A.2.
>
> 6. Sampling means that we sample a z_t from q(z_t|x) and optimizing means that we optimize the function based on $z_{t-1}$ which is sampled from p(z_t|z_{t-1}). This is a Monte Carlo approximation since we sample z_t for multiple times and take every z_t as a fixed value to conditionally sample $z_{t-1}$.
>
> 7. This proposition demonstrates what fine-tuning can do currently. However, different fine-tuning methods do it in different ways. For example, LoRA exactly introduces a new condition c and try to make p(x|c)\approx the true distribution. So we just briefly introduce the goal that one can sample real data sample x from the model.
>
> 8. This is a clerical error. $x^{eval}$ should be x from the dataset. We use these x to evaluate how well the fine-tuned model learns the dataset.
>
> 9. Yes. It is a clerical error.

---

> ### Author Response · Authors · 2023-11-21
> **Response to the review (Cont.)**
>
> Issues with the offset problem and Section 4.1
>
> 1. We agree that the goal can be simply expanded then optimized. That is the main reason for removing our theory. We try to optimize this goal but find that it is empirically weaker than both Eq.16 (named ITA in the new version) and Eq. 15 (named ITA+ in the new version) in the old version of our paper. We do not include the result in the new version of the paper since we remove the introduction of this goal. Our theory cannot explain this phenomenon so that it is removed.
>
> 2. The approximation is about merging the first two terms of the middle line in Eq.13. The sum expectation of $\Vert\delta z_t\Vert^2_2$ approximates $\Vert\mathcal{E}(x)-\mathcal{E}(x’)\Vert^2_2$. However, we rethink the approximation and believe it is not appropriate.
>
> 3. We try to optimize this goal and it performs similarly to the goal we discuss in the first point. Since it does not work well and is not related to our revised method, we do not include the result in the new version of our paper.
>
> 4. We agree that it works very similarly to cosine similarity and thus being unnecessary.
>
> 5. Yes, it does mean magnitude.
>
> Empirical contribution
>
> We understand that one major concern is whether the choice of target or our optimization is the key to our results. We admit that this could be an issue due to our lack of related ablation studies in the first version of our paper. Therefore, we add visualization results of choosing different targets and compare our optimization with the target version of Anti-DB. Results show that both the choice of the target and the optimization goal is the key to our strong empirical results. Please refer to Appendix B.1 for more details.
>
> 1. We agree that targeted attack is the main contribution, which has now been the main topic of our method. We give out experiments on how the contrast and the pattern repetition of the target image impacts the targeted attack and discuss some empirical experience to pick the target image in the Section 3.2 of our new paper. For the recommended ablation studies, (1) The untargeted version of our attack should be Anti-DB, included in our baselines. We provide quantitative comparison and visualization to compare Anti-DB with our attacks. We also give a visual comparison between the targeted version of Ant-DB(please refer to section 3.1 for detailed definition) and our attacks under the same targets(Please refer to Appendix B.1).  As for (2) the prior attacks + targeted, we conduct a case study on it in Appendix A.2. Empirically, it is much weaker than our proposed attacks.
>
> 2. As previously mentioned, Section 3.2 of the new version of our paper now discusses the issues about target selection. Also, Figure 4 shows the visualization results of choosing targets within the domain of natural images. We also give a visualization in Appendix B.3 to give intuitive explanations about the empirical success of our method.

---

> ### Author Response · Authors · 2023-11-21
> **Response to the review (Cont.)**
>
> Questions:
>
> 1. Both share the same approximation. We exchange the sequence of the expectation.
>
> 2. Answered below.
>
> 3. SDEdit conducts minor modification to the image and does not change the structure of the image and only modify details. As shown in Figure 1, all outputs of SDEdit are structurally similar to the original input data. Hence, we use MSSSIM as the structural similarity metric to measure the attack performance. Just as mentioned, we also want to measure the semantic difference (e.g. style difference), thus CLIP-SIM is used to assess the similarity of the images in the latent space of CLIP(trained by aligning text and images). We believe that CLIP-SIM is a good metric to evaluate the high-level semantic distortion of our adversarial attacks. For the same purpose, we use CLIPIQA, a clip-based, non-reference image quality metric to evaluate our text-to-image results. We’ve made a mistake by claiming both CLIP-SIM and MSSSIM both measure the structural differences in our original paper. Thanks for your correction, we’ve updated our explanation of the metrics. Please see “SDEdit & Metrics” under section 5.1 for the updated details.
>
> 4. Yes, it is correct.
>
> 5. They do not share the exact same hyper-parameters, since LoRA training in the attack only lasts for 5 steps and the LoRA training in evaluation lasts for 1000 steps. Although hyper-parameters of the optimizer stay the same, we do not think the result would be quite different. However, the training prompt of LoRA would strongly affect the final performance, especially when it is different from the prompt used in attacking. Therefore, we provide additional experiments to validate our results. First, we conduct experiments where different prompts are used in LoRA training(Please refer to Appendix B.2). The result  shows a degradation in the strength of the attack, but the semantic distortion on the output image is still strong. Second, we give a case of totally black-box evaluation. This case comes from a user from the art community (permission already guaranteed). He/She uses our attack with our default setups (with noise budget 4/255) to protect a batch of his/her paintings and train LoRA upon these paintings. We do not know the details of his/her LoRA training, as well as the model. We take a guess that the model is not Stable Diffusion since Stable Diffusion cannot fit animate style so well. The result shows that our attack performs fairly in this black-box case.
>
> 6. For most common purification methods(e.g. JPEG compression, adding gaussian noise) , we have evaluated them in Section 5.4. Results show our methods are robust to those methods. However, we do not think new potential adaptive purification methods should be discussed in this work. Currently, no specific adaptive purification has been designed to remove the adversarial attacks on LDM. We highly agree with the academic significance of such adaptive purifications. But such purification may help the malicious individual to overcome the adversarial noise and steal copyrighted content from artists and organizations. Since these adversarial attacks on LDM are for social goodness, we do not think releasing such adaptive purification is socially responsible. Such technical attempts should be stopped by rules. Therefore, though we’ve explored some potential methods for pure academic purposes, we would not release them in our paper.

---

> > ### Comment · Reviewer_FuAb · 2023-12-01
> >
> > I really appreciate the author(s)'s willingness to hear my comments and concerns. I took a quick look at the new version of the paper, and I saw a lot of positive changes (in my opinion). However, the main methodology sections (3 and 4) were almost completely rewritten, and the main thesis of the paper as well as the theoretical contributions were completely changed. I feel that this is too large of a change that we should accept in the rebuttal process. So I decided to keep my original assessment.
> >
> > If the final decision is to reject this submission, I would really encourage the author(s) to resubmit their work at the next venue. I do think this is a very interesting and practically impactful problem to study. I would also encourage the author(s) to focus on the empirical aspect of the paper as I believe that it is more suitable for papers in this area.

---

### Author Response · Authors · 2023-11-21
**Updating a major revision of the paper**

We apologize for the very late response, for we have been working for the  major revision after carefully studying the review. The major revision mainly includes two parts:

First, we remove the main body of the theory in the first version. This is because our theory has exactly some mistakes (minor reason) and cannot explain the effectiveness of our proposed method (major reason). We’re sincerely sorry for our mistakes. We’ve been aware of that earlier and started to rethink the unclosed gap from theoretical analysis to empirical results. We believe the core reason is the misalignment between human judges and diffusion-based judges, which are detailedly discussed in Appendix B.2 and Appendix C of our new edition. In the new version of our paper, we mainly discuss the intuition of our proposed methods and give empirical evidence on why they work better than existing methods. We know that this revision may lead to even lower review scores, and are ready to be responsible for that.

Second, we complement extensive experimental results to furthermore verify and explain the empirical effectiveness of our methods. Among them are cross-model transferability, denoising countering, target selection in the proposed targeted attack, black-box evaluation, and so on. We’ve also cross-validated and updated our major experiment result by repeating experiments and taking the mean of the metrics. We hope these results can convince better that our methods outperform existing methods with sound properties in cross-model transferability and denoising countering.

Again, we are sincerely sorry about the theory problem in the first version. As we have done a major revision based on the review, we will really appreciate it if reviewers can read the new version of our paper and give some advice, which must be significant for our further refinement of our paper.

---

### Meta-Review · Area_Chair_4wED · 2023-12-05

**Metareview:**

The paper targets at a better theoretical understanding of existing adversarial attacks on latent diffusion models to preserve data privacy. To fulfill this ambitious goal, the authors formalize different objectives (subgoals) for generating adversarial examples, focusing on misleading latent variables in the forward process, the reverse process, and the finetuning, respectively. These sub-goals unify existing adversarial attacks in one definition. Then, based on this understanding, they propose a joint optimization objectives for these subgoals, as well as a targeted attack method that pulls the adversarial examples towards a targeted image. Experiments show that the proposed strategy achieves clear gains over previous methods on multiple metrics.

The initial version of this paper is rather hard to follow, with many equations and notations that are not fully explained well. Some reviewers raise questions on the novelty and the soundness of the theoretical part. During the rebuttal stage, the authors have submitted a revised version that has substantial changes to the original one. In particular, they basically rewrite the theory and the method part, which diminishes the theoretical contributions. Other reviewers also raise concerns on cherry-picking good results. After an engaging discussion, the reviewers have all agreed that this level of change should deserve another round of review, and they are consistently inclined to reject this paper for this venue. Nevertheless, reviewers do agree that this paper contributes to a better understanding of the design of LDM attacks, so the authors are encouraged to revise the paper according to the detailed comments, and re-submit it to a future venue.

**Justification For Why Not Higher Score:**

The theory of this paper has some soundness problems and it has undergone a significant change during the rebuttal, which should go for another round of review.

**Justification For Why Not Lower Score:**

N/A

---

### Decision · Program_Chairs · 2024-01-16

Reject